# TwinsFormer: Revisiting Inherent Dependencies via Two Interactive Components for Time Series Forecasting

## Abstract

Due to the remarkable ability to capture long-term dependencies, Transformer-based models have shown great potential in time series forecasting. However, real-world time series usually present intricate temporal patterns, making forecasting still challenging in many practical applications. To better grasp inherent dependencies, in this paper, we propose **TwinsFormer**, a novel Transformer-based framework utilizing two interactive components for time series forecasting. Unlike mainstream paradigms that employ plain decomposition, which train the model with two independent branches, we design an interactive strategy centered on the attention module and the feed-forward network to strengthen dependencies through decomposed components. Specifically, we adopt a dual stream approach to facilitate progressive and implicit information interactions for trend and seasonal components. For the seasonal stream, we feed the seasonal component to the attention module and feed-forward network with a subtraction mechanism. Meanwhile, we construct an auxiliary highway (without the attention module) for the trend stream, guided by seasonal signals. In this way, we can avoid the model overlooking inherent dependencies between different components for accurate forecasting. Our interactive strategy, although simple, can be easily adapted as a plug-and-play module to existing Transformer-based methods with minimal additional computational overhead. Experiments on various real-world datasets demonstrate the superiority of TwinsFormer, which outperforms previous state-of-the-art methods in both long-term and short-term forecasting performance.

## 1 Introduction

As a ubiquitous and paramount task in many real-world scenarios (e.g., weather (Wu et al., 2023b), energy (Yin et al., 2021), market (Granger & Newbold, 2014), and transportation (Yin et al., 2021)), time series forecasting has been explored with ongoing passion. Generally, time series forecasting aims to predict future temporal variations based on historical observations of time series, where the primary challenge is how to effectively capture temporal patterns from observed data (Fan et al., 2019; Deng et al., 2021; Shao et al., 2022; Ekambaram et al., 2023; Zhang et al., 2024b). Benefiting from the advancements in deep learning, various representative models with well-designed architectures, such as MLP-based (Wang et al., 2024; Zeng et al., 2023; Li et al., 2023), CNN-based (Wang et al., 2023a; Wu et al., 2023a; Liu et al., 2022a), and Transformer-based (Liu et al., 2024; Zhang & Yan, 2023; Zhou et al., 2022; Piao et al., 2024; Qiu et al., 2025) methods, have been proposed to tackle time series forecasting tasks and demonstrate impressive performance. Since the complex and non-stationary (Liu et al., 2022c; 2025) nature of the real world or systems, the observed series usually involves multitudinous variations, such as increasing, decreasing, and fluctuating, making it still hard to grasp reliable inherent dependencies for accurate forecasting.

To tackle intricate temporal patterns, series decomposition (Robert et al., 1990), which utilizes a moving average kernel to smooth out short-term fluctuations or noise in the time series, has been incorporated into deep models as a basic module. Empowered with various decomposition designs, existing methods (Wu et al., 2021; Zhou et al., 2022; Wang et al., 2023a; Zeng et al., 2023; Stitsyuk & Choi, 2025) generally utilize two independent branches to highlight seasonal and trend properties separately, then combine the seasonal and trend representations for the final prediction. In Figure 1, we visualize the time series and its decomposition components on the ECL dataset. In classic

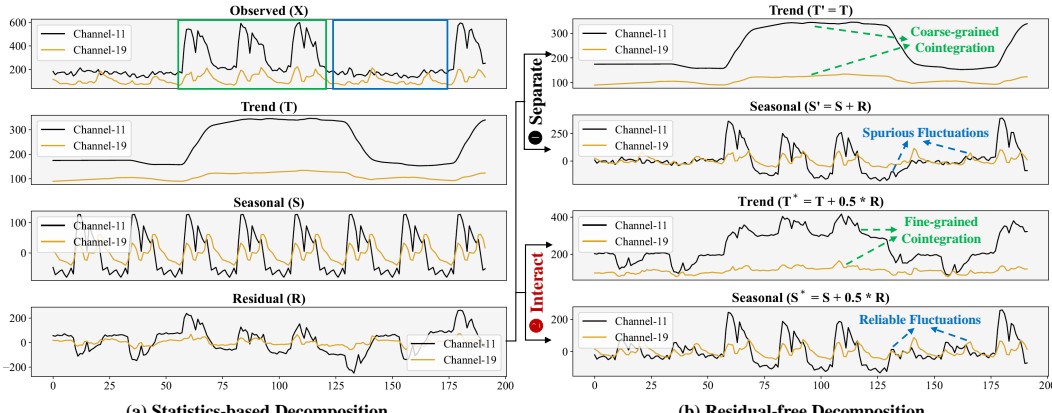

Figure 1: Visualization of different components and their combinations on the ECL dataset. (a) presents the characteristics of different components obtained by STL (Robert et al., 1990), which is a classical statistics-based decomposition (i.e., $X = T + S + R$). Due to the randomness of the Residual ($R$), modern decomposition designs (i.e., ❶) decouple the original time series ($X$) into the Trend ($T$) and Seasonal ($S'$), where $S' = S + R$. To intuitively understand the role of the Residual, we divide $R$ into two equal parts and add them to the Trend and Seasonal for interaction (i.e., ❷).

statistical-based decomposition, time series often contain trend, season, and residual components (i.e., X=T+S+R), which respectively highlight different temporal patterns of the time series. Due to the unpredictability of the residual components, the existing trend-seasonal decompositions ignore the residual components, where the seasonal components obtained by subtracting the trend components from the original time series include the residual components (i.e., S'=S+R). To understand the role of residual components on trend and seasonal components, we divide the residuals into two parts, allowing them to interact with both trend and seasonal components simultaneously. Comparatively, the trend and seasonal components of ❶ maintain worse cointegrations and fluctuations than those of ❷ in Figure 1. Such inconsistencies lead to the learned trend and seasonal representations by independent branches, which may not accurately capture the temporal patterns of the observed series. Therefore, a more rational decomposition design should consider *the interactions between decomposed components to precisely unravel inherent dependencies for observed values.*

To fill this gap, we propose **TwinsFormer**, a Transformer-based framework that explicitly explores inherent dependencies via two interactive components for time series forecasting. **First**, we decompose the observed time series rather than the time series embeddings into trend and seasonal components, to better capture the characteristics of the time series itself. **Second**, since the trend components reflect the long-term fluctuations of the time series, we only feed the seasonal components to the attention and feed-forward modules with a subtraction mechanism to alleviate redundant coding. **Most importantly**, we regard the outputs of the attention and feed-forward modules as supervision information to guide the model to capture the representation of the trend components. With our interactive design, TwinsFormer can successfully aggregate seasonal and trend information to learn inherent dependencies between different components. Experimentally, our proposed TwinsFormer achieves state-of-the-art performance on seven real-world forecasting scenarios, effectively providing an interactive learning scheme for time series forecasting. Our primary contributions are summarized as follows:

- We examine the existing decomposition designs for time series forecasting and find that the interaction between different components is not explored: these designs simply learn separate representations for trend and seasonal components, overlooking non-linear dependencies or significant noise levels among time series.

- We propose TwinsFormer, a Transformer-based framework (to the best of our knowledge, the first) that explicitly explores inherent dependencies by learning implicit and progressive interactions between different components for time series forecasting.

- Extensive experimental results on 13 real-world benchmarks show the superiority of TwinsFormer against state-of-the-art methods. Specifically, TwinsFormer ranks in the top 1 among 11 models on 21 out of 22 average settings, including various prediction lengths and metrics for both long-term and short-term forecasting tasks.

## 2 RELATED WORK

### 2.1 DECOMPOSITIONS FOR TIME SERIES FORECASTING

Due to the capacity of the moving average kernel to smooth out short-term fluctuations or noise in the time series, Autoformer (Wu et al., 2021) initially proposed using the moving average kernel to decompose complex temporal variations into seasonal and trend components. Since then, trend-seasonal decomposition designs based on the moving average kernel have been frequently introduced in time series forecasting works. For instance, SCINet (Liu et al., 2022a) devises a downsample-convolve-interact architecture to extract dynamic temporal features at multiple resolutions with two sub-sequences. DLinear (Zeng et al., 2023) utilizes the series decomposition as the pre-processing before linear regression. MICN (Wang et al., 2023a) adopts multi-scale branches to model the local and global context by decomposing input series into seasonal and trend terms, while TimesNet (Wu et al., 2023a) designs a modular architecture to obtain decomposed components with the Fourier Transform. xPatch (Stitsyuk & Choi, 2025) introduces an exponential trend-seasonal decomposition to assign greater weight to more recent data points while smoothing out older data. More recently, TimeMixer (Wang et al., 2024) mixes multi-scale decomposable components for time series forecasting. Due to the non-linear or non-stationary properties of time series, however, a rudimentary moving averaging kernel may inadequately capture precise trends, which impedes the model from learning inherent dependencies through two independent branches.

### 2.2 TRANSFORMERS FOR TIME SERIES FORECASTING

Transformer-based methods have demonstrated significant success in time series forecasting, primarily because they can effectively model long-term temporal patterns (Li et al., 2019; Zhou et al., 2021; Liu et al., 2022b). However, the self-attention mechanism's quadratic complexity and redundant coding present challenges, leading many existing approaches to modify the attention module to reduce computational overhead. Notable works in this area include Informer (Zhou et al., 2021), which introduces ProbSparse self-attention and distillation techniques, Autoformer (Wu et al., 2021), which incorporates series decomposition with an auto-correlation mechanism, and FEDformer (Zhou et al., 2022), which implements an attention module using a Fourier-enhanced structure. Without modifications to the Transformer, some other attempts focus on the inherent processing of time series, such as stationarity (Liu et al., 2022c; 2023), patching (Du et al., 2023), channel independence (Nie et al., 2023), and inverting operations (Liu et al., 2024), consistently yielding improved performance for time series forecasting. Besides, refurbishing the Transformer in both aspects mentioned above, Crossformer Zhang & Yan (2023) introduces a two-stage attention mechanism and dimension-segment-wise embedding strategy to capture time and variate dependencies. More recently, Fredformer (Piao et al., 2024) employs a frequency-based attention mechanism to mitigate frequency bias, while DUET (Qiu et al., 2025) utilizes dual clustering on the temporal and channel dimensions to enhance forecasting performance.

Building upon the designs in previous works, TwinsFormer introduces an interactive dual-stream architecture that preserves the basic modules of the Transformer. Moreover, we replace the observed series with trend and seasonal components, allowing the model to better learn the inherent dependencies and their interactions. To the best of our knowledge, TwinsFormer is the first attempt to consider interactions between decomposed components on Transformers for time series forecasting.

## 3 TWINSFORMER

**Preliminary.** Given the observation data $X = \{x_1, x_2, \cdots, x_M\} \in \mathbb{R}^{M \times N}$ with $M$ length look-back window and $N$ variates, the goal of multivariate time series forecasting is to predict the future time series $Y = \{x_{M+1}, \cdots, x_{M+\tau}\} \in \mathbb{R}^{\tau \times N}$ at next $\tau$ time steps ($\tau > 1$). Following the idea of decomposition (Robert et al., 1990; Wu et al., 2021), time series can be divided into trend and seasonal components using a moving average kernel. For length-$M$ input series $X \in \mathbb{R}^{M \times N}$, the decomposition process can be formulated as:

$$X_T = AvgPool(Padding(X)),$$
$$X_S = X - X_T, \tag{1}$$

where $X_T$ and $X_S$ are the trend and seasonal components.

Figure 2: Overall framework of TwinsFormer.

## 3.1 STRUCTURE OVERVIEW

Our TwinsFormer, illustrated on the left of Figure 2, adopts the encoder-only architecture, renovating the Transformer to a dual-stream structure with two decomposed components. Before embedding the time series, we decompose the observed series into trend (T) and seasonal (S) components in the channel dimension. Then, we feed seasonal embeddings $E_S$ to the attention module and feed-forward network (FFN) with a subtraction mechanism, while feeding trend embeddings $E_T$ to the interactive module with the supervision of seasonal information (i.e., $A_S$ and $F_s$). Finally, we aggregate seasonal and trend representations for the prediction.

**Embedding the decomposed series as tokens.** For convenience, we denote $X_{m,:}$ as the simultaneously recorded values for all the variates at the $m$ time point, while $X_{:,n}$ as the whole time series of each variate indexed by $n$. Based on Equation 1, the trend and seasonal components of the time series can be formulated as $X_T = AvgPool(Padding(X_{:,n}))$ and $X_S = X - X_T$, respectively. Then, we utilize straightforward linear and dropout layers to create trend and seasonal embeddings with global covariates $X_{mark}$ as follows:

$$
\begin{aligned}
E_T &= Dropout(Linear(Concat(X_T, X_{mark}))), \\
E_S &= Dropout(Linear(Concat(X_S, X_{mark}))).
\end{aligned}
\tag{2}
$$

Note that $Concat(\cdot)$ is used on the dataset containing timestamp information (i.e., $X_{mark}$) and different components (i.e., $X_T$ and $X_S$) through separate linear layers in our experiments. In this way, we map decomposed series data $X_T, X_S \in \mathbb{R}^{N \times M}$ from the original space into a new space, where $E_T, E_S \in \mathbb{R}^{N \times D}$ and $D$ is the embedding dimension.

**Learning interactions with our TwinsBlock.** Unlike existing Transformer variants that attempt to optimize the structure of attention mechanisms in time series forecasting tasks, our TwinsFormer incorporates interactive learning into the Transformer block to explore the interactions between decomposed components, thereby capturing better inherent dependencies.

## 3.2 DUAL-STREAM DESIGN WITH INTERACTIVE MODULE

Keeping the original modules (i.e., the self-attention and feed-forward network (FFN)) of Transformer unchanged, our key design lies in the computationally efficient interactive module, which can guide the model to learn more effective trend and seasonal representations.

**Seasonal Branch.** Since the seasonal components exhibit more fluctuations in the time series data, we feed the seasonal embeddings to the attention and FFN modules to effectively capture the dependencies among the multivariates. Following the attention process of iTransformer (Liu et al., 2024), we regard $E_S \in \mathbb{R}^{N \times D}$ as $N$ $D$-dimension tokens and utilize queries, keys, and values $Q, K, V \in \mathbb{R}^{N \times d_k}$ to obtain the attention-weighted seasonal representations $A_S \in \mathbb{R}^{N \times D}$, where $d_k$ is the projected dimension:

$$
Q = E_s W_1 + b_1, \ \ K = E_s W_2 + b_2, \ \ V = E_s W_3 + b_3, \ \ W_i \in \mathcal{R}^{d_k \times d_k}, \ \ b_i \in \mathcal{R}^{1 \times d_k},
$$

$$
A_S = Softmax(\frac{QK^T}{\sqrt{d_k}})V.
\tag{3}
$$

According to Equation 1, the seasonal components can be regarded as the residual part of the observed time series data. Intuitively, we adopt the idea of residual learning to implement a corrective

strategy by subtracting the outputs of the Attention and FFN modules from the corresponding inputs. The learning process can be formulated as follows:

$$
\begin{aligned}
H_1 &= LayerNorm(E_S - A_S), \\
H_2 &= H_1 - FFN(H_1).
\end{aligned}
\tag{4}
$$

**Trend Branch.** Considering that untrainable moving average kernels lead to unreliable trend patterns, we fuse seasonal information to assist the learning of the trend branch through our interactive module (IM). On the one hand, attention-weighted $A_S$ well reflects the dependencies among multivariate, which can be converted into a coefficient matrix to update the trend embeddings. On the other hand, the signals $F_s$ discarded by the seasonal branch can be regarded as meaningful information to guide the representation of the trend embeddings. Our interactive module is illustrated on the right of Figure 2, and we only use simple structures to train and update the trend branch network:

$$
\begin{aligned}
E_T^1 &= E_T \odot \exp(\alpha(A_s)) + \beta(A_s), \\
E_T^2 &= E_T \odot \exp(\gamma(F_s)) + \mu(F_s),
\end{aligned}
\tag{5}
$$

where $\odot$ denotes element-wise multiplication, $\alpha, \beta, \gamma, \mu$ are four MLPs with ReLU activations, and we obtain the transformed trend $F_T$ by adding $E_T^1$ and $E_T^2$ together.

**Gate Mechanism.** Inspired by the inherent control of cells in RNNs (Zhao et al., 2017; Dey & Salem, 2017), we devise a gate mechanism $\sigma$ at the end of each block for both streams to autonomously regulate the pace of information transmission. The gate mechanism for both seasonal and trend streams can be formulated as:

$$
\begin{aligned}
O_S &= \sigma\left(Conv_1(H_2)\right) \cdot Conv_2(H_2), \\
O_T &= \sigma\left(Conv_3(F_T)\right) \cdot Conv_4(F_T),
\end{aligned}
\tag{6}
$$

where $Conv_1, Conv_2, Conv_3$ and $Conv_4$ are four $1 \times 1$ convolution operations with different parameters. Taking the output of the former TwinsBlock as the input of the latter TwinsBlock, we stack $L$ TwinsBlocks to learn seasonal and trend representations, and then add them together through a linear projection for the ultimate predictive outcomes, i.e., $\{\hat{Y} = Projection(O_S + O_T)\} \in \mathbb{R}^{\tau \times N}$.

### 3.3 THEORETICAL ANALYSIS OF TWINSFORMER

**Structural Constraints.** Given historical time series data $X$, we can obtain its trend and seasonal components (i.e., $X_t$ and $X_s$) by the moving average kernel. For existing time series forecasting methods, we regard the models as $F(\cdot)$, while regarding the independent branches with decomposition designs as $f_t(\cdot)$ and $f_s(\cdot)$, then we can formulate the predictive outputs $\hat{Y}$ as

$$
\hat{Y} = F(f_t(X_t) + f_s(X_s)), \quad where \ X = X_t + X_s.
\tag{7}
$$

Similarly, we define the attention module, FFN, interactive module, and gate mechanism of TwinsFormer as $g(\cdot), h(\cdot), \phi(\cdot),$ and $\sigma$ respectively. Then, the outcomes are:

$$
\hat{Y} = F(\sigma_t(\phi(\overbrace{X_t, g(X_s), h(X_s - g(X_s))}^{\text{interactive learning}}))) \underbrace{\phantom{X_t, g(X_s)}}_{X_t'}
$$
$$
+ \sigma_s(\overbrace{X_s - g(X_s) - h(X_s - g(X_s))}^{\text{residual learning}})),
\underbrace{\phantom{X_s - g(X_s)}}_{X_s'}
\tag{8}
$$

where $\phi(\cdot)$ updates the trend components by Equation 5. By omitting the constraints from various functions on variables, our interactive components can be simplified as

$$
X_s' = X_s - X_1 - X_2, \quad X_t' = X_t + X_1 + X_2.
\tag{9}
$$

Then, the sum of our two interactive components is

$$
\begin{aligned}
X = X_s' + X_t' &= X_s - \cancel{X_1} - \cancel{X_2} + X_t + \cancel{X_1} + \cancel{X_2} \\
&= X_s + X_t.
\end{aligned}
\tag{10}
$$

Based on Equation 10, we can find that our interaction strategy perfectly fits the requirements of the decomposition design without bringing in redundant signals. Furthermore, we can elaborate on the practical implications of our TwinsFormer in mitigating the limitations of the trend-seasonal decomposition. According to Figure 1, observed values contain residual components, which means that the decomposed trend and seasonal components are not completely disentangled. TwinsFormer adopts a dual-stream interaction strategy to implicitly and progressively promote the decoupling of both components by using residual learning and interactive learning. Specifically, we filter out the coupled information (i.e., $X_1$ and $X_2$) from the seasonal

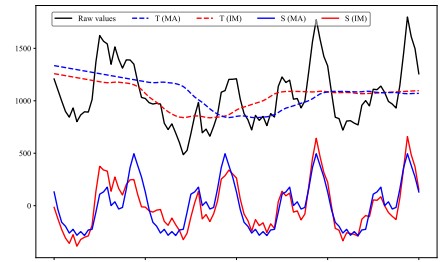

Figure 3: Comparison of the trends (T) and seasonals (S) learned by the existing moving average kernel (MA) and our interactive module (IM).

components and compensate for them with the trend components through transformation mechanisms, allowing us to learn more robust and reliable decomposed representations for accurate time series forecasting. As seen in Figure 3, our interactive module can obtain decomposed components with more consistent variations than the moving average kernel.

**Generalization Bound.** According to the above structural constraints, we can further theoretically justify that such an interactive design makes a tighter generalization error bound. Let $\mathcal{H}_{\text{iTrans}}$ denote the hypothesis space of iTransformer, and $\mathcal{H}_{\text{Twins}}$ denote the hypothesis space of TwinsFormer. Our TwinsFormer enforces a structural decomposition and interaction mechanism that constrains the functions it can represent. Formally, we have the following:

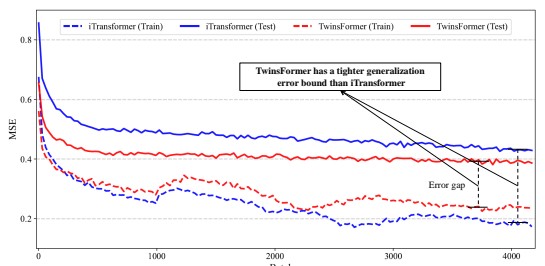

Figure 4: The MSE curve of the models on Traffic.

$$\mathcal{H}_{\text{Twins}} : \mathcal{F}(X_t, X_s) = \{\mathcal{F}(f_t(X_t) + f_s(X_s)) \mid \mathcal{F} \in \mathcal{W}, f_t \in \mathcal{T}, f_s \in \mathcal{S}\},$$
$$\mathcal{H}_{\text{iTrans}} : \mathcal{F}(X) = \{\mathcal{F}(X) \mid \mathcal{F} \in \mathcal{W}, X = X_t + X_s\}, \quad (11)$$

where $\mathcal{W}$ is the family of all possible Transformer-parameterized functions, $\mathcal{T}$ and $\mathcal{S}$ are function families for trend and seasonal components, respectively. Since the decomposition-interaction mechanism can be viewed as imposing a temporal smoothness prior, the hypothesis space of TwinsFormer can be rewritten as:

$$\mathcal{H}_{\text{Twins}} = \{\mathcal{F}(f_t(X_t) + f_s(X_s))\} \approx \{\mathcal{F} \circ (f_t \oplus f_s)(X)\}, \quad (12)$$

where $\circ$ denotes the composition of the function and $\oplus$ denotes the addition of components. This means $\mathcal{H}_{\text{Twins}} \subseteq \mathcal{H}_{\text{iTrans}}$, as it imposes additional structural constraints, where the model must first decompose the input, process the components separately, and then combine them.

According to the properties of Rademacher complexity (Giorgio & Marcello, 2008), for any class of functions $\mathcal{A}$ and $\mathcal{B}$, $\mathfrak{R}(\mathcal{A} \circ \mathcal{B}) \leq \text{Lip}(\mathcal{A}) \cdot \mathfrak{R}(\mathcal{B})$, where $\text{Lip}(\mathcal{A})$ is the Lipschitz constant (Fazlyab et al., 2019) of $\mathcal{A}$. Therefore, we derive the following inequality chain:

$$\mathfrak{R}(\mathcal{H}_{\text{Twins}}) \approx \{\mathcal{F} \circ (\mathcal{T} \oplus \mathcal{S})\} \leq \text{Lip}(\mathcal{F}) \cdot [\mathfrak{R}(\mathcal{T}) + \mathfrak{R}(\mathcal{S})] < \text{Lip}(\mathcal{F}) \cdot \mathfrak{R}(\mathcal{F}) \approx \mathfrak{R}(\mathcal{H}_{\text{iTrans}}). \quad (13)$$

Since Rademacher complexity is a key quantity in deriving upper bounds for generalization error and $\mathfrak{R}(\mathcal{H}_{\text{Twins}}) < \mathfrak{R}(\mathcal{H}_{\text{iTrans}})$, we can find that Twinsformer has a tighter generalization error bound than iTransformer. The detailed analysis is given in Appendix A. To further highlight the generalization of the model, we visualize the MSE curves of TwinsFormer and iTransformer in Figure 4, which indicates that our visualization is consistent with the theoretical analysis.

## 4 EXPERIMENTS

**Benchmarks.** For long-term forecasting, we experiment on 9 public benchmarks, which include ETT (Zhou et al., 2021), ECL (Wu et al., 2021), Exchange (Wu et al., 2021), Traffic (Wu et al., 2021), Weather (Wu et al., 2021) and Solar-energy (Lai et al., 2018) datasets. Moreover, we use PEMS (Liu et al., 2022a) for short-term forecasting.

Table 1: Long-term forecasting results. The lookback length is set to $T = 96$ and all the results are averaged from all predictions $S \in \{96, 192, 336, 720\}$. Avg means further averaged by subsets. A lower MSE or MAE indicates a better forecasting performance.

| Models | TwinsFormer (Ours) | | WPMixer (2025) | | Fredformer (2024) | | iTransformer (2024) | | TimeMixer (2024) | | FilterNet (2024) | | FITS (2024) | | PatchTST (2023) | | DLinear (2023) | | Crossformer (2023) | | TimesNet (2023a) | |
|---|---|---|---|---|---|---|---|---|---|---|---|---|---|---|---|---|---|---|---|---|---|---|
| Metric | MSE | MAE | MSE | MAE | MSE | MAE | MSE | MAE | MSE | MAE | MSE | MAE | MSE | MAE | MSE | MAE | MSE | MAE | MSE | MAE | MSE | MAE |
| ETT (Avg) | **0.365** | **0.384** | 0.372 | 0.390 | 0.368 | 0.387 | 0.383 | 0.399 | 0.381 | 0.396 | 0.381 | 0.398 | 0.402 | 0.404 | 0.381 | 0.397 | 0.442 | 0.444 | 0.685 | 0.578 | 0.391 | 0.404 |
| ECL | **0.163** | **0.253** | 0.166 | 0.255 | 0.176 | 0.269 | 0.178 | 0.270 | 0.183 | 0.272 | 0.205 | 0.290 | 0.384 | 0.434 | 0.205 | 0.290 | 0.212 | 0.300 | 0.244 | 0.334 | 0.192 | 0.295 |
| Exchange | **0.329** | **0.387** | 0.354 | 0.399 | 0.333 | 0.391 | 0.360 | 0.403 | 0.380 | 0.417 | 0.389 | 0.419 | 0.365 | 0.408 | 0.367 | 0.404 | 0.354 | 0.414 | 0.940 | 0.707 | 0.416 | 0.443 |
| Traffic | **0.403** | 0.271 | 0.437 | 0.279 | 0.443 | 0.291 | 0.428 | 0.282 | 0.496 | 0.298 | 0.463 | 0.310 | 0.615 | 0.370 | 0.481 | 0.304 | 0.625 | 0.383 | 0.550 | 0.304 | 0.620 | 0.336 |
| Weather | **0.242** | **0.266** | 0.246 | 0.269 | 0.246 | 0.273 | 0.258 | 0.278 | 0.245 | 0.274 | 0.259 | 0.281 | 0.273 | 0.292 | 0.259 | 0.281 | 0.265 | 0.317 | 0.259 | 0.315 | 0.259 | 0.287 |
| Solar-energy | 0.221 | **0.251** | 0.223 | 0.258 | 0.226 | 0.262 | 0.233 | 0.262 | **0.216** | 0.280 | 0.235 | 0.266 | 0.376 | 0.385 | 0.270 | 0.307 | 0.330 | 0.401 | 0.641 | 0.639 | 0.301 | 0.319 |

**Baselines.** We compare TwinsFormer with 13 representative baselines, including 1) Transformer-based methods: TimeBridge (Liu et al., 2025), TQNet (Lin et al., 2025), Leddam (Yu et al., 2024), Fredformer (Piao et al., 2024) iTransformer (Liu et al., 2024), PatchTST(Nie et al., 2023), Crossformer (Zhang & Yan, 2023); 2) Linear-based methods: WPMixer (Murad et al., 2025), TimeMixer (Wang et al., 2024), DLinear (Zeng et al., 2023), FilterNet (Yi et al., 2024), and FITS (Xu et al., 2024); and 3) TCN-based methods: TimesNet Wu et al. (2023a).

**Implementation details.** All the experiments are implemented in PyTorch (Paszke et al., 2019) and conducted on one NVIDIA 4090 24GB GPU. We use the L2 loss to train the model with the Adam optimizer (Kingma & Ba, 2015), where the training process is stopped early within 30 epochs. Our interactive module can apply to various time series frameworks without introducing any additional hyperparameters. Following iTransformer (Liu et al., 2024), we use the Mean Squared Error (MSE) and Mean Absolute Error (MAE) as the core metrics for the evaluation.

## 4.1 MAIN RESULTS

**Long-term Forecasting.** Comprehensive results for long-term forecasting are presented in Table 1, with the best results highlighted in **bold** and the second-best underlined. TwinsFormer consistently outperforms state-of-the-art models, covering various time series benchmarks with different frequencies, variates, and real-world scenarios. Compared to sophisticated models like WPMixer and Fredformer, TwinsFormer achieves superior performance. Its interactive architecture effectively leverages the inherent trend-seasonal dependencies in time-series data. Specifically, TwinsFormer outperforms WPMixer and Fredformer by a considerable margin, with a $7.5\%$ and $6.9\%$ reduction in MSE among all the datasets for WPMixer and Fredformer, respectively. Although TimeMixer has a subtle reduction in MSE of $0.5\%$ over TwinsFormer in Solar-energy, TwinsFormer achieves lower MAE scores than TimeMixer by $2.9\%$.

**Short-term Forecasting.** TwinsFormer also performs well in short-term forecasting on PEMS datasets. Due to the complex spatiotemporal dependencies among city-wide traffic networks in PEMS benchmarks, many advanced models significantly deteriorate in this task. For instance, TimeMixer employs a multiscale mixing architecture to model complex temporal variations; however, its performance is not as good as that of iTransformer, which simply tokenizes the embedding of time series in the variate dimension. By contrast, TwinsFormer learns the inherent dependencies from the interactions between decomposed components, which can better capture accurate patterns for multivariate time series. As shown in Table 2, TwinsFormer achieves the best performance, confirming the effectiveness of our interactive strategy in modeling complex temporal dynamics.

Table 2: Short-term forecasting results on PEMS datasets. The lookback length is set to $T = 96$ and all the results are averaged from all predictions $S \in \{12, 24, 48, 96\}$.

| Models | TwinsFormer (Ours) | | WPMixer (2025) | | Fredformer (2024) | | iTransformer (2024) | | TimeMixer (2024) | | FilterNet (2024) | | FITS (2024) | | PatchTST (2023) | | DLinear (2023) | | Crossformer (2023) | | TimesNet (2023a) | |
|---|---|---|---|---|---|---|---|---|---|---|---|---|---|---|---|---|---|---|---|---|---|---|
| Metric | MSE | MAE | MSE | MAE | MSE | MAE | MSE | MAE | MSE | MAE | MSE | MAE | MSE | MAE | MSE | MAE | MSE | MAE | MSE | MAE | MSE | MAE |
| PEMS03 | **0.107** | **0.214** | 0.167 | 0.267 | 0.135 | 0.243 | 0.116 | 0.226 | 0.145 | 0.253 | 0.145 | 0.251 | 0.489 | 0.465 | 0.180 | 0.291 | 0.278 | 0.375 | 0.169 | 0.281 | 0.147 | 0.248 |
| PEMS04 | **0.109** | **0.217** | 0.185 | 0.287 | 0.162 | 0.261 | 0.121 | 0.232 | 0.162 | 0.268 | 0.146 | 0.258 | 0.531 | 0.489 | 0.195 | 0.307 | 0.295 | 0.388 | 0.209 | 0.314 | 0.129 | 0.241 |
| PEMS07 | **0.084** | **0.180** | 0.181 | 0.271 | 0.121 | 0.222 | 0.100 | 0.204 | 0.152 | 0.248 | 0.123 | 0.229 | 0.500 | 0.472 | 0.211 | 0.303 | 0.329 | 0.395 | 0.235 | 0.315 | 0.124 | 0.225 |
| PEMS08 | **0.122** | **0.211** | 0.226 | 0.299 | 0.161 | 0.250 | 0.151 | 0.234 | 0.209 | 0.296 | 0.172 | 0.260 | 0.534 | 0.487 | 0.280 | 0.321 | 0.379 | 0.416 | 0.268 | 0.307 | 0.193 | 0.271 |
| Avg | **0.106** | **0.206** | 0.190 | 0.281 | 0.145 | 0.244 | 0.122 | 0.224 | 0.167 | 0.266 | 0.147 | 0.250 | 0.514 | 0.478 | 0.217 | 0.306 | 0.320 | 0.394 | 0.220 | 0.304 | 0.148 | 0.246 |

## 4.2 ABLATION STUDIES

To verify the effectiveness of each main component of TwinsFormer, we provide indispensable ablation studies for every possible design on decomposition and interactions. Specifically, we disable or replace certain designs as model variants and experiment on two long-term (i.e., ECL and Traffic) and two short-term forecasting (i.e., PEMS03 and PEMS07) datasets. As shown in Table 3, we conduct an insightful analysis of decomposition and interactions based on the following observation.

Table 3: Ablation studies for TwinsFormer. We disable or replace each component of both decomposition and interactions over four datasets. ✓ and ✗ indicate with and without certain components, respectively. The average results of all predicted lengths are listed here.

| Design | Decomposition | − | $F_T$ | $A_S$ | $F_S$ | $\sigma$ | ECL MSE | ECL MAE | Traffic MSE | Traffic MAE | PEMS03 MSE | PEMS03 MAE | PEMS07 MSE | PEMS07 MAE |
|---|---|---|---|---|---|---|---|---|---|---|---|---|---|---|
| **TwinsFormer** | ✓ | ✓ | ✓ | ✓ | ✓ | ✓ | **0.163** | **0.253** | **0.403** | **0.271** | **0.107** | **0.214** | **0.084** | **0.180** |
| ① | ✗ | ✓ | ✓ | ✓ | ✓ | ✓ | 0.178 | 0.275 | 0.413 | 0.288 | 0.126 | 0.236 | 0.112 | 0.208 |
| ② | swap | ✓ | ✓ | ✓ | ✓ | ✓ | 0.165 | 0.258 | 0.406 | 0.276 | 0.110 | 0.218 | 0.091 | 0.184 |
| ③ | ✓ | + | ✓ | ✓ | ✓ | ✓ | 0.179 | 0.271 | 0.421 | 0.283 | 0.115 | 0.224 | 0.106 | 0.197 |
| ④ | ✓ | ✓ | ✗ | ✓ | ✓ | ✓ | 0.177 | 0.272 | 0.410 | 0.285 | 0.123 | 0.232 | 0.105 | 0.200 |
| ⑤ | ✓ | ✓ | ✓ | ✗ | ✓ | ✓ | 0.172 | 0.267 | 0.408 | 0.278 | 0.118 | 0.219 | 0.093 | 0.188 |
| ⑥ | ✓ | ✓ | ✓ | ✓ | ✗ | ✓ | 0.166 | 0.261 | 0.408 | 0.277 | 0.111 | 0.220 | 0.098 | 0.192 |
| ⑦ | ✓ | ✓ | ✓ | ✓ | ✓ | ✗ | 0.175 | 0.269 | 0.424 | 0.283 | 0.119 | 0.232 | 0.117 | 0.205 |

**Ablation on decomposition.** Considering that the trend and seasonal components in the decomposition design are fed to different network branches, we disable the decomposition by using two original observed series as inputs (i.e., ①) and swap trend and seasonal components (i.e., ②) for ablation analysis. In ablation ① and ②, we observe significant decreases in forecasting performance for both long-term and short-term predictions, which demonstrates that our integration of the decomposition into the Transformer architecture is reasonable and effective.

**Ablation on interactions.** For the interactions, we verify effectiveness by gradually removing or replacing components. In ablation ③, we replace the subtraction mechanism (i.e., −) with original addition skip connections (i.e., +), and the results on ③ show a decline in forecasting accuracy. This illustrates that decomposed components can better satisfy the requirements of the Transformer architecture by using the subtraction mechanism. Meanwhile, the results in ③ further highlight the rationality of the decomposition design, which is consistent with the theoretical analysis in Section 3.3. In ablations ④, ⑤, ⑥, and ⑦, we eliminate the impact of $F_T$, $A_S$, $F_S$, and gate mechanism $\sigma$ for interactive learning, respectively. These four ablations all result in significant drops in forecasting performance, indicating that all inputs for interactive learning can effectively enhance the performance of TwinsFormer. The above observations highlight the substantial influence of our strategy, which utilizes residual and interactive learning in the Transformer architecture.

## 4.3 MODEL ANALYSIS

**Compatibility Study.** To verify the compatibility and promoting effect of our framework, we adopt different decomposition initializations and apply the interactive module (IM) to three excel-

Table 4: Compatibility study for TwinsFormer. We adopt the moving average kernel (MA), Fourier-based transformation (FB), and learnable decomposition (LD) initializations and integrate our interactive module (IM) into three Transformer-based forecasters.

| Model | | TwinsFormer (Ours) | | | | | | TimeBridge (2025) | | | | TQNet (2025) | | | | Leddam (2024) | | | |
|---|---|---|---|---|---|---|---|---|---|---|---|---|---|---|---|---|---|---|---|
| Setup | | MA | | LD | | FD | | Original | | + IM | | Original | | + IM | | Original | | + IM | |
| Metric | | MSE | MAE | MSE | MAE | MSE | MAE | MSE | MAE | MSE | MAE | MSE | MAE | MSE | MAE | MSE | MAE | MSE | MAE |
| ECL | 96 | **0.134** | **0.223** | 0.136 | 0.228 | 0.135 | 0.226 | 0.120 | 0.214 | **0.115** | **0.210** | 0.134 | 0.229 | **0.132** | **0.228** | 0.141 | 0.235 | **0.138** | **0.233** |
| | 192 | **0.154** | **0.240** | 0.155 | 0.242 | 0.156 | 0.243 | 0.142 | 0.237 | **0.138** | **0.235** | 0.154 | 0.247 | **0.150** | **0.242** | 0.159 | 0.252 | **0.158** | **0.252** |
| | 336 | **0.165** | **0.257** | 0.163 | 0.254 | 0.167 | 0.258 | 0.156 | 0.252 | **0.154** | **0.251** | 0.169 | 0.264 | **0.163** | **0.261** | 0.173 | 0.268 | **0.172** | **0.267** |
| | 720 | **0.198** | **0.290** | 0.200 | 0.293 | 0.201 | 0.293 | 0.179 | 0.278 | **0.178** | **0.276** | 0.201 | 0.294 | **0.199** | **0.292** | 0.201 | 0.295 | **0.200** | **0.293** |
| | Avg | **0.163** | **0.253** | 0.164 | 0.254 | 0.165 | 0.255 | 0.149 | 0.245 | **0.146** | **0.243** | 0.164 | 0.259 | **0.161** | **0.256** | 0.169 | 0.263 | **0.167** | **0.261** |
| Traffic | 96 | 0.379 | **0.258** | 0.381 | 0.260 | **0.378** | 0.259 | 0.340 | 0.240 | **0.338** | **0.239** | 0.413 | 0.261 | **0.406** | **0.258** | 0.426 | 0.276 | **0.416** | **0.265** |
| | 192 | **0.388** | **0.265** | 0.387 | 0.266 | 0.390 | 0.268 | 0.343 | 0.250 | **0.340** | **0.248** | 0.432 | 0.271 | **0.418** | **0.267** | 0.458 | 0.289 | **0.429** | **0.272** |
| | 336 | **0.407** | **0.272** | 0.410 | 0.275 | 0.411 | 0.279 | 0.363 | 0.257 | **0.361** | **0.255** | 0.450 | 0.277 | **0.433** | **0.272** | 0.486 | 0.297 | **0.435** | **0.284** |
| | 720 | 0.439 | 0.289 | 0.442 | 0.292 | **0.438** | **0.286** | 0.393 | 0.271 | **0.392** | **0.270** | 0.486 | 0.295 | **0.448** | **0.295** | 0.498 | 0.313 | **0.452** | **0.292** |
| | Avg | **0.403** | **0.271** | 0.405 | 0.273 | 0.404 | 0.273 | 0.360 | 0.255 | **0.358** | **0.253** | 0.445 | 0.276 | **0.426** | **0.273** | 0.467 | 0.294 | **0.433** | **0.278** |

lent Transformer-based forecasters. On the one hand, we replace the moving average (MA) with frequency-based (FB) or learnable decomposition (LD) strategies. On the other hand, we integrate our interactive design into the TimeBridge (Liu et al., 2025), TQNet (Lin et al., 2025), and Leddam (Yu et al., 2024) without modifying their hyperparameters. As seen in Table 4, the average performance gap achieved by TwinsFormer does not exceed 0.02 under different decomposition initializations, which illustrates the favorable decomposition compatibility of TwinsFormer. Moreover, our interactive technique consistently improves the original baselines' performance, indicating its portability and superiority across different Trenasformer-based architectures.

**Dependency Study.** To provide an intuitive understanding of the learned representations by our dual-stream framework, we visualize the multivariate correlations in Figure 5. It can be observed that the multivariate correlations learned by iTransformer are redundant compared to the ground truth. In contrast, feeding the attention branch with seasonal components can better capture multivariate correlations than feeding it with trend components, which is consistent with the results of ② in Table 3. Those observations indicate that our interactive design can learn more accurate dependencies than iTransformer and achieve better forecasting performance.

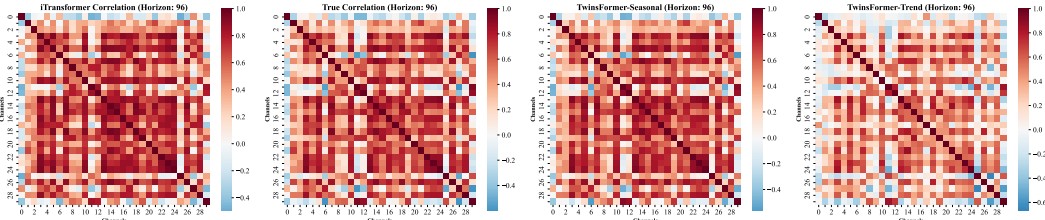

Figure 5: Analysis of multivariate correlations on ECL. Zoom in for more details.

**Lookback Sensitivity.** As argued in (Zeng et al., 2023) and (Nie et al., 2023), most of the Transformer-based models will not improve the forecasting performance with an increasing lookback length due to the distracted attention on the longer input (Liu et al., 2024). However, our TwinsFormer reduces the MSE scores with enlarged historical information available, which is consistent with the theoretical analysis using statistical methods (Box & Jenkins, 1968). As seen in Figure 6, the forecasting results keep improving in most cases where the prediction length $S$ belongs to $\{96, 192, 336, 720\}$ as the receptive field increases. These improvements confirm that our TwinsFormer can effectively capture inherent dependencies from a longer lookback window.

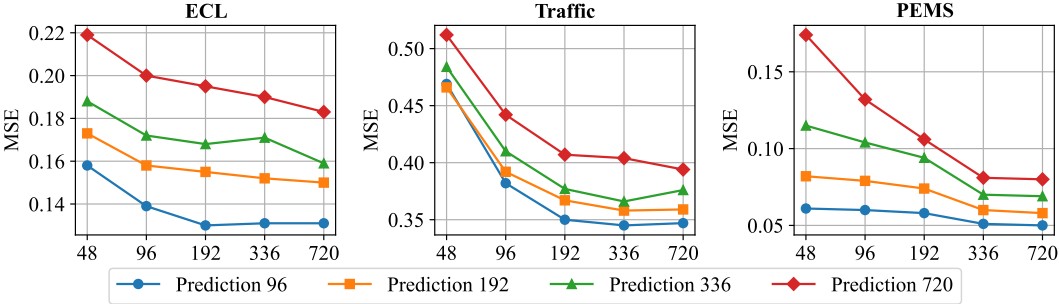

Figure 6: Forecasting performance with different lookback lengths on three datasets.

## 5 CONCLUSION AND FUTURE WORK

Leveraging the strengths of decomposition for mining temporal patterns and attention for capturing multivariate correlations, we propose TwinsFormer, which models inherent dependencies in time series through two interactive branches. Empowered by a novel interactive design, TwinsFormer seamlessly integrates decomposition into the Transformer architecture, enabling effective learning of time series representations. Experiments show that TwinsFormer achieves state-of-the-art performance on both long-term and short-term forecasting tasks. Detailed visualizations, ablation studies, and analyses further demonstrate the effectiveness and generalization of our framework. For future work, we plan to explore more efficient interaction designs for non-Transformer architectures and extend our evaluation to a broader range of time series tasks.

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

## A   THEORETICAL FOUNDATION FOR GENERALIZATION BOUNDS

Time series forecasting presents unique challenges due to the inherent temporal dependencies and non-stationary characteristics of sequential data. TwinsFormer addresses these challenges through explicit decomposition of the input series $X = \{x_1, x_2, \ldots, x_T\}$ into trend ($X_t$) and seasonal ($X_s$) components, with $X = X_t + X_s$. Its predictive mechanism incorporates specialized interactive modules, which can be defined as follows:

$$Y = \mathcal{F}\left(\sigma_t(\phi(X_t, g(X_s), h(X_s - g(X_s))) + \sigma_s(X_s - g(X_s) - h(X_s - g(X_s))))\right), \quad (14)$$

where $g(\cdot), h(\cdot), \phi(\cdot)$ represent attention, FFN, and interaction modules, and $\sigma$ denotes gating mechanisms. The simplified interaction ensures information preservation:

$$X'_s = X_s - X_1 - X_2, \quad X'_t = X_t + X_1 + X_2, \quad X = X'_s + X'_t. \quad (15)$$

This structural design imposes temporal-aware regularization, which we theoretically show leads to tighter generalization bounds in time series settings.

### A.1   PROBLEM FORMULATION AND DEFINITIONS

**Definition A.1 (Time Series Generation Process)** *Let $\{Z_t\}_{t=1}^{\infty}$ be a stochastic process representing the underlying time series, where each $Z_t = (X_t, Y_t)$ consists of input features $X_t \in \mathcal{X}$ and target values $Y_t \in \mathcal{Y}$. The data-generating process follows:*

$$Z_t = f(Z_{t-1}, Z_{t-2}, \ldots, Z_{t-p}, \epsilon_t), \quad (16)$$

*where $p$ is the order of temporal dependence and $\epsilon_t$ represents innovation noise.*

**Definition A.2 (Time Series Forecasting Task)** *For forecasting with lookback window $L$ and forecast horizon $H$, we define training samples as:*

$$D = \left\{\left(X^{(i)}, Y^{(i)}\right)\right\}_{i=1}^{m}, \quad (17)$$

*where each sample is constructed from consecutive time points:*

$$\begin{aligned}
X^{(i)} &= \{z_{t-L}, z_{t-L+1}, \ldots, z_{t-1}\}, \\
Y^{(i)} &= \{z_t, z_{t+1}, \ldots, z_{t+H-1}\}.
\end{aligned} \quad (18)$$

The samples exhibit inherent temporal dependence: $\left(X^{(i)}, Y^{(i)}\right)$ and $\left(X^{(i+1)}, Y^{(i+1)}\right)$ are highly correlated. We assume the time series process is stationary and satisfies the $\beta$-mixing condition.

**Definition A.3 ($\beta$-mixing Coefficient)** *The $\beta$-mixing coefficient of the process $\mathbf{Z}$ is defined as:*

$$\beta(k) = \sup_t \mathbb{E}\left[\sup_{A \in \mathcal{F}_{t+k}^{\infty}} \left|\mathbb{P}(A|\mathcal{F}_{-\infty}^t) - \mathbb{P}(A)\right|\right], \quad (19)$$

*where $\mathcal{F}_a^b$ is the $\sigma$-algebra generated by $(Z_a, \ldots, Z_b)$. The process is said to be $\beta$-mixing if $\beta(k) \to 0$ as $k \to \infty$. We assume an exponential decay of dependence: $\beta(k) \leq \beta_0 e^{-\lambda k}$ for some $\beta_0, \lambda > 0$.*

### A.2   HYPOTHESIS SPACES AND THEIR COMPLEXITIES

Let $\mathcal{H}_{\text{Trans}}$ denote the hypothesis space of a standard Transformer model. The TwinsFormer hypothesis class $\mathcal{H}_{\text{Twins}}$ is a subset of $\mathcal{H}_{\text{Trans}}$ with an inductive bias for temporal decomposition:

$$\mathcal{H}_{\text{Twins}} = \{h : h(X) = \mathcal{F}(f_t(X_t) + f_s(X_s)) \mid X = X_t + X_s, f_t \in \mathcal{T}, f_s \in \mathcal{S}, \mathcal{F} \in \mathcal{W}\}, \quad (20)$$

where $\mathcal{T}$ and $\mathcal{S}$ are the function classes for trend and seasonal components, respectively, and $\mathcal{W}$ is the class of final projection functions. The functions $f_t$ and $f_s$ are realized by the dedicated attention and FFN modules in TwinsFormer.

This constraint is particularly effective for time series as it aligns with the inherent trend-seasonality decomposition of temporal processes, reducing the effective hypothesis space.

### A.3 GENERALIZATION BOUNDS FOR $\beta$-MIXING PROCESSES

For dependent data, the Rademacher complexity is adapted to account for temporal structure. For a $\beta$-mixing process with mixing coefficient $\beta(k)$, the following generalization bound holds with probability at least $1 - \delta$ for all $h \in \mathcal{H}$ (Mohri & Rostamizadeh, 2012):

$$L_{\mathcal{D}}(h) \leq \hat{L}_S(h) + 2\mathfrak{R}_m^{\text{TS}}(\mathcal{H}) + M\sqrt{\frac{2\log(1/\delta)}{m}} + M\beta\left(\lfloor m/2 \rfloor\right), \tag{21}$$

where $M$ is a bound on the loss function, and $\mathfrak{R}_m^{\text{TS}}(\mathcal{H})$ is the time-series Rademacher complexity.

The time-series Rademacher complexity for a $\beta$-mixing process can be bounded by:

$$\mathfrak{R}_m^{\text{TS}}(\mathcal{H}) \leq \inf_{\epsilon > 0}\left\{2\epsilon + 3\sqrt{\frac{2\log(2/\epsilon)}{m}}\right\} + C\sqrt{\frac{\beta^{-1}(1/m)}{m}}, \tag{22}$$

where $C$ is a constant depending on the function class complexity.

### A.4 COMPLEXITY REDUCTION THROUGH DECOMPOSITION

The key insight is that the decomposition operation effectively reduces the complexity of the function class. Instead of having a single complex function $\mathcal{H}_{\text{iTrans}} \in \mathcal{H}_{\text{Trans}}$ learn the entire mapping, where $\mathcal{H}_{\text{iTrans}}$ is a representative Transformer-based model, TwinsFormer employs a decomposition $X = X_t + X_s$ followed by two specialized functions $f_t$ and $f_s$.

Assuming the combination function $\mathcal{F}$ is Lipschitz continuous with constant $\text{Lip}(\mathcal{F})$, and leveraging the fact that the Rademacher complexity of a sum of function classes is bounded by the sum of their complexities (Bartlett & Mendelson, 2002), we have:

$$\mathfrak{R}(\mathcal{H}_{\text{Twins}}) \leq \text{Lip}(\mathcal{F}) \cdot \left(\mathfrak{R}(\mathcal{T}) + \mathfrak{R}(\mathcal{S})\right). \tag{23}$$

The standard Transformer hypothesis class $\mathcal{H}_{\text{iTrans}}$ can be viewed as learning the combined function directly, i.e., $\mathcal{H}_{\text{iTrans}} \approx \{\mathcal{G}(X)\}$ where $\mathcal{G}$ is a highly complex function. Crucially, for the same level of empirical performance, the decomposition prior in TwinsFormer implies that the sum $\mathfrak{R}(\mathcal{T}) + \mathfrak{R}(\mathcal{S})$ is smaller than the complexity required for a monolithic function $\mathfrak{R}(\mathcal{G})$ to achieve the same decomposition effect implicitly. Therefore, we conclude:

$$\mathfrak{R}_m^{\text{TS}}(\mathcal{H}_{\text{Twins}}) \leq \text{Lip}(\mathcal{F}) \cdot \left(\mathfrak{R}_m^{\text{TS}}(\mathcal{T}) + \mathfrak{R}_m^{\text{TS}}(\mathcal{S})\right) < \mathfrak{R}_m^{\text{TS}}(\mathcal{H}_{\text{iTrans}}). \tag{24}$$

This inequality holds because the structural prior of TwinsFormer allows it to use simpler functions to achieve the same goal, thus reducing effective complexity.

### A.5 TIGHTER GENERALIZATION BOUND FOR TWINSFORMER

Substituting the complexity reduction into the time-series generalization bound yields:

$$\begin{aligned}
L_{\mathcal{D}}(h_{\text{Twins}}) &\leq \hat{L}_S(h_{\text{Twins}}) + 2\mathfrak{R}_m^{\text{TS}}(\mathcal{H}_{\text{Twins}}) + M\sqrt{\frac{2\log(1/\delta)}{m}} + M\beta\left(\lfloor m/2 \rfloor\right) \\
&< \hat{L}_S(h_{\text{Twins}}) + 2\mathfrak{R}_m^{\text{TS}}(\mathcal{H}_{\text{iTrans}}) + M\sqrt{\frac{2\log(1/\delta)}{m}} + M\beta\left(\lfloor m/2 \rfloor\right).
\end{aligned} \tag{25}$$

Although the second line of the bound uses $\mathfrak{R}_m^{\text{TS}}(\mathcal{H}_{\text{Trans}})$, the key point is that for TwinsFormer, the *effective* complexity term $\mathfrak{R}_m^{\text{TS}}(\mathcal{H}_{\text{Twins}})$ is significantly smaller than $\mathfrak{R}_m^{\text{TS}}(\mathcal{H}_{\text{Trans}})$. This leads to a tighter *actual* generalization bound for TwinsFormer when the empirical risk $\hat{L}_S$ is similar. The structural prior of explicit decomposition translates into a reduced complexity measure and, consequently, improved generalization guarantees in time-series forecasting.

Given that the interactive module is designed as a structural constraint with negligible computational complexity, TwinsFormer achieves this superior generalization performance without substantially increasing the computational burden.

## B    IMPLEMENTATION DETAILS

**Benchmarks details.** We evaluate the performance of TwinsFormer compared with various baselines on 13 well-established benchmarks [1], which are detailed in Table 5.

**Metrics details.** Regarding evaluation metrics, we utilize the mean square error (MSE) and mean absolute error (MAE) for long-term and short-term forecasting:

$$\text{MSE} = \frac{1}{L}\sum_{i=1}^{L}(X_i - \hat{X}_i)^2, \qquad \text{MAE} = \sum_{i=1}^{L}|X_i - \hat{X}_i|,$$

where $X, \hat{X} \in \mathbb{R}^{L \times N}$ denote the ground truth and prediction results for $N$ variates in the future $L$ time steps. $|\cdot|$ means the absolute value operation.

Table 5: Detailed descriptions of benchmarks. Channel denotes the number of variates in each dataset. The prediction length indicates four prediction settings. The dataset size is split into (Train, Validation, Test). Frequency denotes the sampling interval of time points.

| Tasks | Benchmarks | Channels | Prediction Length | Dataset Size | Frequency | Information |
|---|---|---|---|---|---|---|
| Long-term Forecasting | ETTm1 | 7 | {96, 192, 336, 720} | (34465, 11521, 11521) | 15min | Electricity |
| | ETTm2 | 7 | | (34465, 11521, 11521) | 15min | Electricity |
| | ETTh1 | 7 | | (8545, 2881, 2881) | Hourly | Electricity |
| | ETTh2 | 7 | | (8545, 2881, 2881) | Hourly | Electricity |
| | ECL | 321 | | (18317, 2633, 5261) | Hourly | Electricity |
| | Traffic | 862 | | (12185, 1757, 3509) | Hourly | Transportation |
| | Exchange | 8 | | (5120, 665, 1422) | Daily | Economy |
| | Weather | 21 | | (36792, 5271, 10540) | 10min | Weather |
| | Solar-energy | 137 | | (36601, 5161, 10417) | 10min | Electricity |
| Short-term Forecasting | PEMS03 | 358 | {12, 24, 48, 96} | (15617, 5135, 5135) | 5min | Transportation |
| | PEMS04 | 307 | | (10172, 3375, 3375) | 5min | Transportation |
| | PEMS07 | 883 | | (16911, 5622, 5622) | 5min | Transportation |
| | PEMS08 | 170 | | (10690, 3548, 3548) | 5min | Transportation |

**Algorithm details.** We provide the pseudo-code of TwinsFormer in Algorithm 1.

---

**Algorithm 1** Workflow of our TwinsFormer.

---

**Input:** Input lookback time series $X \in \mathbb{R}^{T \times N}$; Input length $T$, prediction length $L$, and variates number $N$; Token dimension $D$, TwinsBlock number $M$, and moving average kernel size $k$.

**Output:** The prediction results $\hat{X} \in \mathbb{R}^{L \times N}$.

1: ▷ Using the moving average kernel and padding operations to decompose time series.

2: $X_T = AvgPool(Padding(X)), X_S = X - X_T$       ▷ $X_T, X_S \in \mathbb{R}^{T \times N}$

3: ▷ Embedding series into variate tokens by Multi-layer Perceptron.

4: $E_T^0 = \text{Embed}_T(X_T.transpose), E_S^0 = \text{Embed}_S(X_S.transpose)$     ▷ $E_T^0, E_S^0 \in \mathbb{R}^{N \times D}$

5: **for** $m$ in $\{1, \cdots, M\}$ **do**

6:     ▷ Self-attention mechanism and feed-forward network are applied for the seasonal branch.

7:     $E_S^{m-1} = \text{LayerNorm}(E_S^{m-1} - \text{Attn}(E_S^{m-1}))$       ▷ $E_S^{m-1} \in \mathbb{R}^{N \times D}$

8:     $E_S^m = E_S^{m-1} - \text{FFN}(E_S^{m-1})$       ▷ $E_S^m \in \mathbb{R}^{N \times D}$

9:     ▷ Interactive module (IM) is implemented with four MLPs (i.e., $\alpha, \beta, \gamma, \mu$).

10:     $E_T^{m-1,1} = E_T^{m-1} \odot \exp \alpha(\text{Attn}(E_S^{m-1})) + \beta(\text{Attn}(E_S^{m-1}))$    ▷ $E_T^{m-1,1} \in \mathbb{R}^{N \times D}$

11:     $E_T^{m-1,2} = E_T^{m-1} \odot \exp \gamma(\text{FFN}(E_S^{m-1})) + \mu(\text{FFN}(E_S^{m-1}))$    ▷ $E_T^{m-1,2} \in \mathbb{R}^{N \times D}$

12:     ▷ Adding gate mechanism to seasonal and trend branches.

13:     $E_S^m = LayerNorm(Sigmoid(Conv(E_S^m)) * E_S^m)$       ▷ $E_S^m \in \mathbb{R}^{N \times D}$

14:     $E_T^m = Sigmoid(Conv(E_T^{m-1,1} + E_T^{m-1,2})) * (E_T^{m-1,1} + E_T^{m-1,2})$    ▷ $E_T^m \in \mathbb{R}^{N \times D}$

15: **end for**

16: $\hat{X} = Projector(E_S^m + E_T^m)$       ▷ $\hat{X} \in \mathbb{R}^{N \times L}$

17: $\hat{X} = \hat{X}.transpose$       ▷ $\hat{X} \in \mathbb{R}^{L \times N}$

18: **return** $\hat{X}$

---

[1] All the datasets are publicly available at `https://github.com/thuml/iTransformer`

## C  Full Main Results

Due to the space limitation, we provide the full multivariate forecasting results here. Specifically, Table 6 contains the detailed results of all prediction lengths on 9 well-acknowledged benchmarks for long-term forecasting, while Table 7 includes the full short-term forecasting results on 4 challenging citywide traffic datasets.

Table 6: Full results of the long-term forecasting task. We compare extensive competitive models under different prediction lengths $S \in \{96, 192, 336, 720\}$. The input sequence length is set to 96 for all baselines. Avg means the average results from all four prediction lengths.

| Models | TwinsFormer (Ours) | | WPMixer (2025) | | Fredformer (2024) | | iTransformer (2024) | | TimeMixer (2024) | | FilterNet (2024) | | FITS (2024) | | PatchTST (2023) | | DLinear (2023) | | Crossformer (2023) | | TimesNet (2023a) | |
|---|---|---|---|---|---|---|---|---|---|---|---|---|---|---|---|---|---|---|---|---|---|---|
| Metric | MSE | MAE | MSE | MAE | MSE | MAE | MSE | MAE | MSE | MAE | MSE | MAE | MSE | MAE | MSE | MAE | MSE | MAE | MSE | MAE | MSE | MAE |
| ETTm1 96 | 0.315 | 0.354 | 0.326 | 0.362 | 0.331 | 0.368 | 0.334 | 0.368 | 0.320 | 0.355 | 0.327 | 0.372 | 0.355 | 0.376 | 0.329 | 0.367 | 0.345 | 0.372 | 0.404 | 0.426 | 0.338 | 0.375 |
| ETTm1 192 | 0.362 | 0.384 | 0.372 | 0.394 | 0.365 | 0.389 | 0.377 | 0.391 | 0.362 | 0.382 | 0.367 | 0.387 | 0.486 | 0.445 | 0.367 | 0.385 | 0.380 | 0.389 | 0.540 | 0.451 | 0.374 | 0.387 |
| ETTm1 336 | 0.396 | 0.402 | 0.402 | 0.405 | 0.405 | 0.413 | 0.426 | 0.420 | 0.396 | 0.406 | 0.409 | 0.414 | 0.531 | 0.475 | 0.399 | 0.410 | 0.413 | 0.413 | 0.532 | 0.515 | 0.410 | 0.411 |
| ETTm1 720 | 0.457 | 0.439 | 0.476 | 0.453 | 0.463 | 0.448 | 0.491 | 0.459 | 0.458 | 0.445 | 0.477 | 0.452 | 0.600 | 0.513 | 0.454 | 0.439 | 0.474 | 0.453 | 0.666 | 0.589 | 0.478 | 0.450 |
| ETTm1 Avg | 0.383 | 0.395 | 0.394 | 0.404 | 0.391 | 0.405 | 0.407 | 0.410 | 0.384 | 0.397 | 0.395 | 0.406 | 0.493 | 0.452 | 0.387 | 0.400 | 0.403 | 0.407 | 0.513 | 0.496 | 0.400 | 0.406 |
| ETTm2 96 | 0.169 | 0.251 | 0.182 | 0.263 | 0.177 | 0.259 | 0.180 | 0.264 | 0.176 | 0.259 | 0.175 | 0.258 | 0.182 | 0.266 | 0.175 | 0.259 | 0.193 | 0.292 | 0.287 | 0.366 | 0.187 | 0.267 |
| ETTm2 192 | 0.236 | 0.289 | 0.238 | 0.294 | 0.243 | 0.301 | 0.250 | 0.309 | 0.242 | 0.303 | 0.240 | 0.301 | 0.253 | 0.312 | 0.241 | 0.302 | 0.284 | 0.362 | 0.414 | 0.492 | 0.249 | 0.309 |
| ETTm2 336 | 0.292 | 0.330 | 0.306 | 0.342 | 0.302 | 0.340 | 0.311 | 0.348 | 0.303 | 0.339 | 0.311 | 0.347 | 0.313 | 0.349 | 0.305 | 0.343 | 0.369 | 0.427 | 0.597 | 0.542 | 0.321 | 0.351 |
| ETTm2 720 | 0.397 | 0.397 | 0.409 | 0.407 | 0.397 | 0.396 | 0.412 | 0.407 | 0.396 | 0.399 | 0.414 | 0.405 | 0.416 | 0.406 | 0.402 | 0.400 | 0.554 | 0.522 | 1.730 | 1.042 | 0.408 | 0.403 |
| ETTm2 Avg | 0.274 | 0.317 | 0.284 | 0.327 | 0.280 | 0.324 | 0.288 | 0.332 | 0.279 | 0.325 | 0.285 | 0.328 | 0.391 | 0.333 | 0.281 | 0.326 | 0.350 | 0.401 | 0.757 | 0.610 | 0.291 | 0.333 |
| ETTh1 96 | 0.375 | 0.391 | 0.377 | 0.394 | 0.373 | 0.392 | 0.386 | 0.405 | 0.384 | 0.400 | 0.388 | 0.410 | 0.386 | 0.395 | 0.414 | 0.419 | 0.386 | 0.400 | 0.423 | 0.448 | 0.384 | 0.402 |
| ETTh1 192 | 0.439 | 0.431 | 0.434 | 0.426 | 0.433 | 0.420 | 0.441 | 0.436 | 0.437 | 0.429 | 0.442 | 0.449 | 0.437 | 0.424 | 0.460 | 0.445 | 0.437 | 0.432 | 0.471 | 0.474 | 0.436 | 0.429 |
| ETTh1 336 | 0.469 | 0.436 | 0.466 | 0.443 | 0.470 | 0.437 | 0.487 | 0.458 | 0.472 | 0.446 | 0.491 | 0.456 | 0.476 | 0.446 | 0.501 | 0.466 | 0.481 | 0.459 | 0.570 | 0.546 | 0.491 | 0.469 |
| ETTh1 720 | 0.473 | 0.472 | 0.471 | 0.470 | 0.467 | 0.456 | 0.503 | 0.491 | 0.586 | 0.531 | 0.505 | 0.493 | 0.484 | 0.470 | 0.500 | 0.488 | 0.519 | 0.516 | 0.653 | 0.621 | 0.521 | 0.500 |
| ETTh1 Avg | 0.439 | 0.433 | 0.437 | 0.433 | 0.436 | 0.426 | 0.454 | 0.447 | 0.470 | 0.451 | 0.457 | 0.452 | 0.446 | 0.434 | 0.469 | 0.454 | 0.456 | 0.452 | 0.529 | 0.522 | 0.458 | 0.450 |
| ETTh2 96 | 0.285 | 0.332 | 0.287 | 0.336 | 0.293 | 0.342 | 0.297 | 0.349 | 0.297 | 0.348 | 0.293 | 0.343 | 0.294 | 0.340 | 0.302 | 0.348 | 0.333 | 0.387 | 0.745 | 0.584 | 0.340 | 0.374 |
| ETTh2 192 | 0.364 | 0.385 | 0.365 | 0.383 | 0.371 | 0.389 | 0.380 | 0.400 | 0.369 | 0.392 | 0.379 | 0.396 | 0.377 | 0.391 | 0.388 | 0.400 | 0.477 | 0.476 | 0.877 | 0.656 | 0.402 | 0.414 |
| ETTh2 336 | 0.397 | 0.419 | 0.418 | 0.422 | 0.382 | 0.409 | 0.428 | 0.432 | 0.427 | 0.435 | 0.419 | 0.430 | 0.416 | 0.425 | 0.426 | 0.433 | 0.594 | 0.541 | 1.104 | 0.763 | 0.452 | 0.452 |
| ETTh2 720 | 0.406 | 0.430 | 0.423 | 0.441 | 0.415 | 0.434 | 0.427 | 0.445 | 0.462 | 0.463 | 0.449 | 0.460 | 0.418 | 0.437 | 0.431 | 0.446 | 0.831 | 0.657 | 1.104 | 0.763 | 0.462 | 0.468 |
| ETTh2 Avg | 0.363 | 0.392 | 0.373 | 0.396 | 0.365 | 0.394 | 0.383 | 0.407 | 0.389 | 0.409 | 0.385 | 0.407 | 0.376 | 0.398 | 0.387 | 0.407 | 0.559 | 0.515 | 0.942 | 0.684 | 0.414 | 0.427 |
| ECL 96 | 0.134 | 0.223 | 0.135 | 0.225 | 0.147 | 0.241 | 0.148 | 0.240 | 0.153 | 0.244 | 0.147 | 0.245 | 0.198 | 0.274 | 0.181 | 0.281 | 0.197 | 0.282 | 0.219 | 0.314 | 0.168 | 0.272 |
| ECL 192 | 0.154 | 0.240 | 0.159 | 0.242 | 0.165 | 0.258 | 0.162 | 0.253 | 0.168 | 0.259 | 0.160 | 0.254 | 0.363 | 0.422 | 0.188 | 0.274 | 0.196 | 0.285 | 0.231 | 0.322 | 0.184 | 0.289 |
| ECL 336 | 0.165 | 0.257 | 0.168 | 0.259 | 0.177 | 0.273 | 0.178 | 0.269 | 0.185 | 0.275 | 0.173 | 0.283 | 0.444 | 0.490 | 0.204 | 0.293 | 0.209 | 0.301 | 0.246 | 0.337 | 0.198 | 0.300 |
| ECL 720 | 0.198 | 0.290 | 0.201 | 0.295 | 0.213 | 0.304 | 0.225 | 0.317 | 0.227 | 0.312 | 0.210 | 0.309 | 0.532 | 0.551 | 0.246 | 0.324 | 0.245 | 0.333 | 0.280 | 0.363 | 0.220 | 0.320 |
| ECL Avg | 0.163 | 0.253 | 0.166 | 0.255 | 0.176 | 0.269 | 0.178 | 0.270 | 0.183 | 0.272 | 0.205 | 0.290 | 0.384 | 0.434 | 0.205 | 0.290 | 0.212 | 0.300 | 0.244 | 0.334 | 0.192 | 0.295 |
| Exchange 96 | 0.079 | 0.198 | 0.083 | 0.201 | 0.084 | 0.202 | 0.086 | 0.206 | 0.099 | 0.218 | 0.091 | 0.211 | 0.087 | 0.208 | 0.088 | 0.205 | 0.088 | 0.218 | 0.256 | 0.367 | 0.107 | 0.234 |
| Exchange 192 | 0.170 | 0.293 | 0.174 | 0.296 | 0.178 | 0.302 | 0.177 | 0.299 | 0.196 | 0.313 | 0.186 | 0.305 | 0.185 | 0.306 | 0.176 | 0.299 | 0.176 | 0.315 | 0.470 | 0.509 | 0.226 | 0.344 |
| Exchange 336 | 0.317 | 0.402 | 0.325 | 0.412 | 0.319 | 0.408 | 0.331 | 0.417 | 0.359 | 0.432 | 0.380 | 0.449 | 0.343 | 0.425 | 0.301 | 0.397 | 0.313 | 0.427 | 1.268 | 0.883 | 0.367 | 0.448 |
| Exchange 720 | 0.749 | 0.653 | 0.833 | 0.687 | 0.749 | 0.651 | 0.847 | 0.691 | 0.864 | 0.703 | 0.898 | 0.712 | 0.846 | 0.694 | 0.901 | 0.714 | 0.839 | 0.695 | 1.767 | 1.068 | 0.964 | 0.746 |
| Exchange Avg | 0.329 | 0.387 | 0.354 | 0.399 | 0.333 | 0.391 | 0.360 | 0.403 | 0.380 | 0.417 | 0.389 | 0.419 | 0.365 | 0.408 | 0.367 | 0.404 | 0.354 | 0.414 | 0.940 | 0.707 | 0.416 | 0.443 |
| Traffic 96 | 0.379 | 0.258 | 0.396 | 0.266 | 0.406 | 0.277 | 0.395 | 0.268 | 0.473 | 0.287 | 0.430 | 0.294 | 0.601 | 0.361 | 0.462 | 0.295 | 0.650 | 0.396 | 0.522 | 0.290 | 0.593 | 0.321 |
| Traffic 192 | 0.388 | 0.265 | 0.427 | 0.274 | 0.426 | 0.290 | 0.417 | 0.276 | 0.486 | 0.294 | 0.452 | 0.307 | 0.603 | 0.365 | 0.466 | 0.296 | 0.598 | 0.370 | 0.530 | 0.293 | 0.617 | 0.336 |
| Traffic 336 | 0.407 | 0.272 | 0.444 | 0.281 | 0.437 | 0.292 | 0.433 | 0.283 | 0.488 | 0.298 | 0.470 | 0.316 | 0.609 | 0.366 | 0.482 | 0.304 | 0.605 | 0.373 | 0.558 | 0.305 | 0.629 | 0.336 |
| Traffic 720 | 0.439 | 0.289 | 0.480 | 0.294 | 0.462 | 0.305 | 0.467 | 0.302 | 0.536 | 0.314 | 0.498 | 0.323 | 0.648 | 0.387 | 0.514 | 0.322 | 0.645 | 0.394 | 0.589 | 0.328 | 0.640 | 0.350 |
| Traffic Avg | 0.403 | 0.271 | 0.437 | 0.279 | 0.433 | 0.291 | 0.428 | 0.282 | 0.496 | 0.298 | 0.463 | 0.310 | 0.615 | 0.370 | 0.481 | 0.304 | 0.625 | 0.383 | 0.550 | 0.304 | 0.620 | 0.336 |
| Weather 96 | 0.158 | 0.199 | 0.164 | 0.204 | 0.163 | 0.207 | 0.174 | 0.214 | 0.163 | 0.209 | 0.162 | 0.207 | 0.196 | 0.236 | 0.177 | 0.218 | 0.196 | 0.255 | 0.158 | 0.230 | 0.172 | 0.220 |
| Weather 192 | 0.207 | 0.243 | 0.212 | 0.246 | 0.211 | 0.251 | 0.221 | 0.254 | 0.209 | 0.252 | 0.215 | 0.252 | 0.240 | 0.271 | 0.225 | 0.259 | 0.237 | 0.296 | 0.206 | 0.277 | 0.219 | 0.261 |
| Weather 336 | 0.263 | 0.285 | 0.268 | 0.287 | 0.267 | 0.292 | 0.278 | 0.296 | 0.264 | 0.293 | 0.273 | 0.295 | 0.292 | 0.307 | 0.278 | 0.297 | 0.283 | 0.335 | 0.272 | 0.335 | 0.280 | 0.306 |
| Weather 720 | 0.339 | 0.336 | 0.341 | 0.339 | 0.343 | 0.341 | 0.358 | 0.347 | 0.345 | 0.345 | 0.351 | 0.346 | 0.365 | 0.354 | 0.354 | 0.348 | 0.345 | 0.381 | 0.398 | 0.418 | 0.365 | 0.359 |
| Weather Avg | 0.242 | 0.266 | 0.246 | 0.269 | 0.246 | 0.273 | 0.258 | 0.278 | 0.245 | 0.274 | 0.259 | 0.281 | 0.273 | 0.292 | 0.259 | 0.281 | 0.265 | 0.317 | 0.259 | 0.315 | 0.259 | 0.287 |
| Solar-Energy 96 | 0.188 | 0.222 | 0.189 | 0.237 | 0.185 | 0.233 | 0.203 | 0.237 | 0.189 | 0.259 | 0.205 | 0.242 | 0.319 | 0.353 | 0.234 | 0.286 | 0.290 | 0.378 | 0.310 | 0.331 | 0.250 | 0.292 |
| Solar-Energy 192 | 0.219 | 0.246 | 0.223 | 0.248 | 0.227 | 0.253 | 0.233 | 0.261 | 0.222 | 0.283 | 0.233 | 0.265 | 0.367 | 0.387 | 0.267 | 0.310 | 0.320 | 0.398 | 0.734 | 0.725 | 0.296 | 0.318 |
| Solar-Energy 336 | 0.240 | 0.265 | 0.239 | 0.273 | 0.246 | 0.284 | 0.248 | 0.273 | 0.231 | 0.292 | 0.249 | 0.278 | 0.408 | 0.403 | 0.290 | 0.315 | 0.353 | 0.415 | 0.750 | 0.735 | 0.319 | 0.330 |
| Solar-Energy 720 | 0.236 | 0.269 | 0.241 | 0.275 | 0.247 | 0.276 | 0.249 | 0.275 | 0.231 | 0.292 | 0.353 | 0.281 | 0.411 | 0.395 | 0.289 | 0.317 | 0.356 | 0.413 | 0.769 | 0.765 | 0.338 | 0.337 |
| Solar-Energy Avg | 0.221 | 0.251 | 0.223 | 0.258 | 0.226 | 0.262 | 0.233 | 0.262 | 0.216 | 0.280 | 0.235 | 0.266 | 0.376 | 0.385 | 0.270 | 0.307 | 0.330 | 0.401 | 0.641 | 0.639 | 0.301 | 0.319 |
| 1st Count | 32 | 37 | 1 | 1 | 7 | 6 | 0 | 0 | 6 | 1 | 0 | 0 | 0 | 0 | 0 | 0 | 0 | 0 | 0 | 0 | 0 | 0 |

TwinsFormer achieves the best forecasting performance among 11 models on various prediction horizons for both long-term and short-term forecasting tasks. To be concrete, TwinsFormer outperforms all baselines on 69 out of the 90 settings, including various prediction lengths and metrics across 9 long-term benchmarks. Meanwhile, TwinsFormer outperforms all baselines on all settings of varying prediction lengths and metrics across four short-term datasets.

Table 7: Full results of the short-term forecasting task. We compare extensive competitive models under different prediction lengths $S \in \{12, 24, 48, 96\}$. The input sequence length is set to 96 for all baselines. Avg means the average results from all four prediction lengths.

| Models | | TwinsFormer (Ours) | | WPMixer (2025) | | Fredformer (2024) | | iTransformer (2024) | | TimeMixer (2024) | | FilterNet (2024) | | FITS (2024) | | PatchTST (2023) | | DLinear (2023) | | Crossformer (2023) | | TimesNet (2023a) | |
|---|---|---|---|---|---|---|---|---|---|---|---|---|---|---|---|---|---|---|---|---|---|---|---|
| Metric | | MSE | MAE | MSE | MAE | MSE | MAE | MSE | MAE | MSE | MAE | MSE | MAE | MSE | MAE | MSE | MAE | MSE | MAE | MSE | MAE | MSE | MAE |
| **PEMS03** 12 | | **0.063** | **0.165** | 0.076 | 0.188 | 0.068 | 0.174 | 0.069 | 0.175 | 0.077 | 0.187 | 0.071 | 0.177 | 0.117 | 0.226 | 0.099 | 0.216 | 0.122 | 0.243 | 0.090 | 0.203 | 0.085 | 0.192 |
| 24 | | **0.084** | **0.192** | 0.113 | 0.226 | 0.093 | 0.202 | 0.097 | 0.208 | 0.112 | 0.224 | 0.102 | 0.213 | 0.235 | 0.324 | 0.142 | 0.259 | 0.201 | 0.317 | 0.121 | 0.240 | 0.118 | 0.223 |
| 48 | | **0.119** | **0.231** | 0.191 | 0.292 | 0.146 | 0.258 | 0.131 | 0.243 | 0.169 | 0.277 | 0.162 | 0.272 | 0.541 | 0.521 | 0.211 | 0.319 | 0.333 | 0.425 | 0.202 | 0.317 | 0.155 | 0.260 |
| 96 | | **0.161** | **0.267** | 0.288 | 0.363 | 0.228 | 0.330 | 0.168 | 0.279 | 0.220 | 0.322 | 0.244 | 0.340 | 1.062 | 0.790 | 0.269 | 0.370 | 0.457 | 0.515 | 0.262 | 0.367 | 0.228 | 0.317 |
| Avg | | **0.107** | **0.214** | 0.167 | 0.267 | 0.135 | 0.243 | 0.116 | 0.226 | 0.145 | 0.253 | 0.145 | 0.251 | 0.489 | 0.465 | 0.180 | 0.291 | 0.278 | 0.375 | 0.169 | 0.281 | 0.147 | 0.248 |
| **PEMS04** 12 | | **0.072** | **0.179** | 0.092 | 0.204 | 0.085 | 0.189 | 0.081 | 0.188 | 0.092 | 0.203 | 0.082 | 0.190 | 0.129 | 0.239 | 0.105 | 0.224 | 0.148 | 0.272 | 0.098 | 0.218 | 0.087 | 0.195 |
| 24 | | **0.093** | **0.201** | 0.128 | 0.243 | 0.117 | 0.224 | 0.099 | 0.211 | 0.127 | 0.239 | 0.110 | 0.224 | 0.246 | 0.337 | 0.153 | 0.275 | 0.224 | 0.340 | 0.131 | 0.256 | 0.103 | 0.215 |
| 48 | | **0.121** | **0.228** | 0.213 | 0.315 | 0.174 | 0.276 | 0.133 | 0.247 | 0.188 | 0.294 | 0.160 | 0.276 | 0.568 | 0.539 | 0.229 | 0.339 | 0.355 | 0.437 | 0.205 | 0.326 | 0.136 | 0.250 |
| 96 | | **0.148** | **0.260** | 0.307 | 0.384 | 0.273 | 0.354 | 0.172 | 0.283 | 0.240 | 0.337 | 0.234 | 0.343 | 1.181 | 0.843 | 0.291 | 0.389 | 0.452 | 0.504 | 0.402 | 0.457 | 0.190 | 0.303 |
| Avg | | **0.109** | **0.217** | 0.185 | 0.287 | 0.162 | 0.261 | 0.121 | 0.232 | 0.162 | 0.268 | 0.146 | 0.258 | 0.531 | 0.489 | 0.195 | 0.307 | 0.295 | 0.388 | 0.209 | 0.314 | 0.129 | 0.241 |
| **PEMS07** 12 | | **0.055** | **0.145** | 0.073 | 0.184 | 0.063 | 0.158 | 0.067 | 0.167 | 0.069 | 0.172 | 0.064 | 0.163 | 0.109 | 0.222 | 0.095 | 0.207 | 0.115 | 0.242 | 0.094 | 0.200 | 0.082 | 0.181 |
| 24 | | **0.070** | **0.164** | 0.111 | 0.219 | 0.089 | 0.192 | 0.086 | 0.189 | 0.106 | 0.212 | 0.093 | 0.200 | 0.230 | 0.327 | 0.150 | 0.262 | 0.210 | 0.329 | 0.139 | 0.247 | 0.101 | 0.204 |
| 48 | | **0.094** | **0.192** | 0.237 | 0.328 | 0.136 | 0.241 | 0.110 | 0.214 | 0.185 | 0.282 | 0.137 | 0.248 | 0.551 | 0.531 | 0.253 | 0.340 | 0.398 | 0.458 | 0.311 | 0.369 | 0.134 | 0.238 |
| 96 | | **0.117** | **0.217** | 0.303 | 0.254 | 0.197 | 0.298 | 0.138 | 0.244 | 0.246 | 0.327 | 0.198 | 0.306 | 1.112 | 0.809 | 0.346 | 0.404 | 0.594 | 0.553 | 0.396 | 0.442 | 0.181 | 0.279 |
| Avg | | **0.084** | **0.180** | 0.181 | 0.271 | 0.121 | 0.222 | 0.100 | 0.204 | 0.152 | 0.248 | 0.123 | 0.229 | 0.500 | 0.472 | 0.211 | 0.303 | 0.329 | 0.395 | 0.235 | 0.315 | 0.124 | 0.225 |
| **PEMS08** 12 | | **0.071** | **0.171** | 0.091 | 0.201 | 0.081 | 0.185 | 0.080 | 0.183 | 0.097 | 0.205 | 0.080 | 0.182 | 0.122 | 0.233 | 0.168 | 0.232 | 0.154 | 0.276 | 0.165 | 0.214 | 0.112 | 0.212 |
| 24 | | **0.091** | **0.189** | 0.137 | 0.246 | 0.112 | 0.214 | 0.118 | 0.221 | 0.156 | 0.262 | 0.114 | 0.219 | 0.236 | 0.330 | 0.224 | 0.281 | 0.248 | 0.353 | 0.215 | 0.260 | 0.141 | 0.238 |
| 48 | | **0.128** | **0.219** | 0.265 | 0.343 | 0.174 | 0.267 | 0.186 | 0.265 | 0.269 | 0.345 | 0.184 | 0.284 | 0.562 | 0.540 | 0.321 | 0.354 | 0.440 | 0.470 | 0.315 | 0.355 | 0.198 | 0.283 |
| 96 | | **0.198** | **0.266** | 0.410 | 0.407 | 0.277 | 0.335 | 0.221 | 0.267 | 0.313 | 0.373 | 0.309 | 0.356 | 1.216 | 0.846 | 0.408 | 0.417 | 0.674 | 0.565 | 0.377 | 0.397 | 0.320 | 0.351 |
| Avg | | **0.122** | **0.211** | 0.226 | 0.299 | 0.161 | 0.250 | 0.151 | 0.234 | 0.209 | 0.296 | 0.172 | 0.260 | 0.534 | 0.487 | 0.280 | 0.321 | 0.379 | 0.416 | 0.268 | 0.307 | 0.193 | 0.271 |
| 1st Count | | **20** | **20** | 0 | 0 | 0 | 0 | 0 | 0 | 0 | 0 | 0 | 0 | 0 | 0 | 0 | 0 | 0 | 0 | 0 | 0 | 0 | 0 |

## D  ERROR BARS

We obtain the standard deviation of TwinsFormer performance by training the model with 5 different random seeds over 12 datasets. As seen in Table 8, the error bars of all the results are tiny, which exhibits that the performance of TwinsFormer is robust and reliable.

Table 8: Robustness of TwinsFormer performance obtained from 5 random seeds on 12 benchmarks.

| Dataset | ETTm1 | | ETTm2 | | ETTh2 | | ECL | |
|---|---|---|---|---|---|---|---|---|
| Metrics | MSE | MAE | MSE | MAE | MSE | MAE | MSE | MAE |
| 96 | $0.315 \pm 0.001$ | $0.354 \pm 0.001$ | $0.169 \pm 0.001$ | $0.251 \pm 0.001$ | $0.285 \pm 0.001$ | $0.332 \pm 0.001$ | $0.134 \pm 0.002$ | $0.223 \pm 0.001$ |
| 192 | $0.362 \pm 0.001$ | $0.384 \pm 0.002$ | $0.236 \pm 0.001$ | $0.289 \pm 0.001$ | $0.364 \pm 0.002$ | $0.385 \pm 0.003$ | $0.154 \pm 0.001$ | $0.240 \pm 0.001$ |
| 336 | $0.396 \pm 0.002$ | $0.402 \pm 0.003$ | $0.292 \pm 0.001$ | $0.330 \pm 0.002$ | $0.397 \pm 0.004$ | $0.419 \pm 0.002$ | $0.165 \pm 0.002$ | $0.257 \pm 0.002$ |
| 720 | $0.457 \pm 0.002$ | $0.439 \pm 0.003$ | $0.397 \pm 0.002$ | $0.397 \pm 0.001$ | $0.406 \pm 0.003$ | $0.430 \pm 0.001$ | $0.198 \pm 0.002$ | $0.290 \pm 0.003$ |

| Dataset | Traffic | | Exchange | | Solar-Energy | | Weather | |
|---|---|---|---|---|---|---|---|---|
| Metrics | MSE | MAE | MSE | MAE | MSE | MAE | MSE | MAE |
| 96 | $0.379 \pm 0.001$ | $0.258 \pm 0.000$ | $0.079 \pm 0.001$ | $0.198 \pm 0.001$ | $0.188 \pm 0.002$ | $0.222 \pm 0.002$ | $0.158 \pm 0.001$ | $0.199 \pm 0.002$ |
| 192 | $0.388 \pm 0.001$ | $0.265 \pm 0.002$ | $0.170 \pm 0.002$ | $0.293 \pm 0.002$ | $0.219 \pm 0.002$ | $0.246 \pm 0.002$ | $0.207 \pm 0.001$ | $0.243 \pm 0.002$ |
| 336 | $0.407 \pm 0.003$ | $0.272 \pm 0.002$ | $0.317 \pm 0.002$ | $0.402 \pm 0.001$ | $0.240 \pm 0.001$ | $0.265 \pm 0.002$ | $0.263 \pm 0.002$ | $0.285 \pm 0.002$ |
| 720 | $0.439 \pm 0.001$ | $0.289 \pm 0.002$ | $0.749 \pm 0.012$ | $0.653 \pm 0.004$ | $0.236 \pm 0.002$ | $0.269 \pm 0.001$ | $0.339 \pm 0.001$ | $0.336 \pm 0.001$ |

| Dataset | PEMS03 | | PEMS04 | | PEMS07 | | PEMS08 | |
|---|---|---|---|---|---|---|---|---|
| Metrics | MSE | MAE | MSE | MAE | MSE | MAE | MSE | MAE |
| 12 | $0.063 \pm 0.001$ | $0.165 \pm 0.001$ | $0.072 \pm 0.001$ | $0.179 \pm 0.001$ | $0.055 \pm 0.002$ | $0.145 \pm 0.001$ | $0.071 \pm 0.001$ | $0.171 \pm 0.001$ |
| 24 | $0.084 \pm 0.001$ | $0.192 \pm 0.001$ | $0.093 \pm 0.002$ | $0.201 \pm 0.001$ | $0.070 \pm 0.001$ | $0.164 \pm 0.002$ | $0.091 \pm 0.002$ | $0.189 \pm 0.001$ |
| 48 | $0.119 \pm 0.001$ | $0.231 \pm 0.002$ | $0.121 \pm 0.001$ | $0.228 \pm 0.001$ | $0.094 \pm 0.002$ | $0.192 \pm 0.001$ | $0.128 \pm 0.002$ | $0.219 \pm 0.001$ |
| 96 | $0.161 \pm 0.002$ | $0.267 \pm 0.002$ | $0.148 \pm 0.002$ | $0.260 \pm 0.001$ | $0.117 \pm 0.002$ | $0.217 \pm 0.001$ | $0.198 \pm 0.002$ | $0.266 \pm 0.001$ |

## E  MORE EXTRA RESULTS

**Efficiency analysis.** Our TwinsFormer is a Transformer-based architecture with dual-stream interactions, where the trend branch is composed of linear layers and sigmoid activation functions. Therefore, like other Transformer models, the main complexity of TwinsFormer is $O(N^2)$, which comes from the seasonal branch with the attention module. Note that the $N$ for TwinsFormer is related to the number of variates, while the $N$ for most Transformer-based models is affected by the

lookback length. Furthermore, the efficiency of TwinsFormer surpasses that of most Transformer variants in datasets with a relatively small number of variates (i.e., $N < 96$). By learning multivariate correlations through variate tokens following iTransformer (Liu et al., 2024), TwinsFormer can consistently exhibit superior computational efficiency on high-dimensional channel datasets.

**Hyperparameter Sensitivity.** We evaluate the hyperparameter sensitivity of TwinsFormer in terms of the learning rate, the number of Twinsblock, and the hidden dimension of variate tokens. As shown in Figure 7, the performance fluctuates under different hyperparameter settings. We can observe that the learning rate, as the most common hyperparameter, should be carefully selected for different datasets. In most cases, increasing the number of Twinsblock tends to strengthen the model performance, especially in datasets with numerous varieties. For scenarios involving many attributes, the forecasting performance decreases when the hidden dimension of variate tokens exceeds 1024.

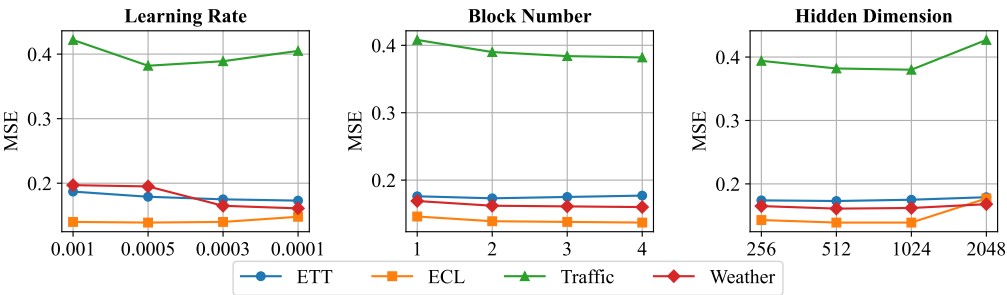

Figure 7: Hyperparameter sensitivity concerning the learning rate, the number of Twinsblock, and the hidden dimension of variate tokens. The results are recorded with an input length of $T = 96$ and a prediction length of $S = 96$ on four benchmarks.

**Showcases.** We present supplementary forecasting showcases in the Traffic dataset, comparing them with five representative models. As seen in 8, TwinsFormer exhibits superior forecasting performance with the most precise future series variations.

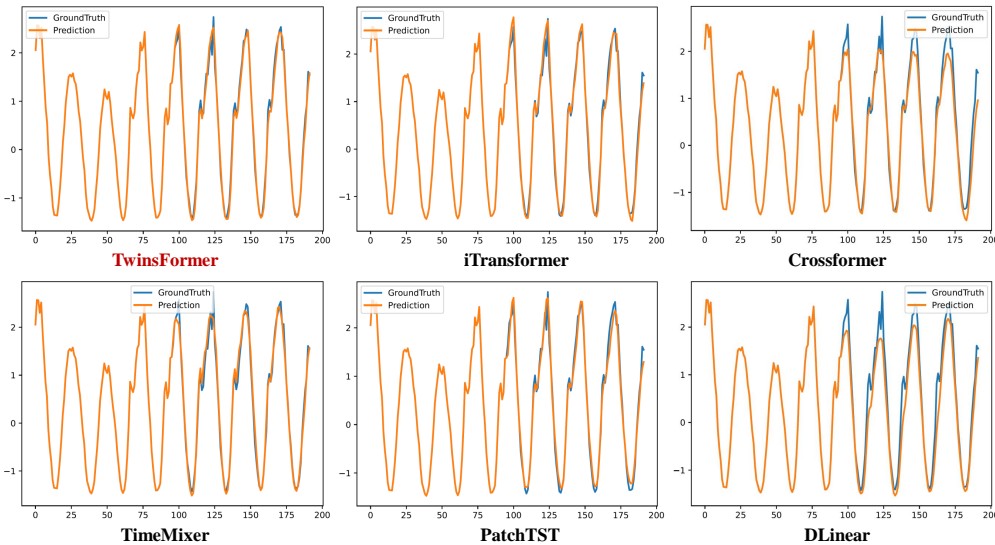

Figure 8: Traffic prediction cases among different models under the input-96-predict-96 setting.

Table 9: Full results on more benchmarks under different settings, where input-96-predict-$\{96, 192, 336, 720\}$ is used for Wind, ZafNoo, and CzeLan, while input-36-predict-$\{24, 36, 48, 60\}$ is used for Covid-19 and Wiki. Avg means the average results from all four prediction lengths.

| Models | | TwinsFormer (Ours) | | WPMixer (2025) | | Fredformer (2024) | | iTransformer (2024) | | TimeMixer (2024) | | FilterNet (2024) | | PatchTST (2023) | | Dlinear (2023) | |
|---|---|---|---|---|---|---|---|---|---|---|---|---|---|---|---|---|---|
| Metrics | | MSE | MAE | MSE | MAE | MSE | MAE | MSE | MAE | MSE | MAE | MSE | MAE | MSE | MAE | MSE | MAE |
| Wind | 96 | 0.825 | 0.608 | 0.884 | 0.642 | 0.921 | 0.655 | 0.848 | 0.625 | 0.855 | 0.620 | 0.953 | 0.674 | 0.889 | 0.652 | 0.881 | 0.632 |
| | 192 | 0.974 | 0.672 | 0.992 | 0.712 | 1.078 | 0.748 | 1.028 | 0.692 | 1.032 | 0.712 | 1.147 | 0.764 | 1.076 | 0.769 | 1.034 | 0.715 |
| | 336 | 1.069 | 0.747 | 1.173 | 0.802 | 1.215 | 0.819 | 1.150 | 0.776 | 1.153 | 0.776 | 1.311 | 0.825 | 1.209 | 0.809 | 1.159 | 0.779 |
| | 720 | 1.183 | 0.804 | 1.402 | 0.891 | 1.323 | 0.858 | 1.245 | 0.829 | 1.233 | 0.809 | 1.323 | 0.864 | 1.304 | 0.851 | 1.233 | 0.815 |
| | avg | 1.013 | 0.708 | 1.113 | 0.762 | 1.134 | 0.770 | 1.068 | 0.731 | 1.068 | 0.729 | 1.184 | 0.782 | 1.120 | 0.770 | 1.077 | 0.735 |
| ZafNoo | 96 | 0.422 | 0.403 | 0.442 | 0.426 | 0.434 | 0.428 | 0.432 | 0.411 | 0.432 | 0.419 | 0.541 | 0.473 | 0.444 | 0.426 | 0.434 | 0.411 |
| | 192 | 0.472 | 0.438 | 0.525 | 0.469 | 0.498 | 0.456 | 0.487 | 0.448 | 0.479 | 0.449 | 0.708 | 0.575 | 0.498 | 0.456 | 0.484 | 0.444 |
| | 336 | 0.506 | 0.456 | 0.578 | 0.509 | 0.530 | 0.480 | 0.521 | 0.469 | 0.521 | 0.469 | 0.851 | 0.661 | 0.530 | 0.480 | 0.518 | 0.464 |
| | 720 | 0.542 | 0.476 | 0.624 | 0.581 | 0.574 | 0.499 | 0.553 | 0.491 | 0.543 | 0.483 | 0.876 | 0.699 | 0.574 | 0.499 | 0.548 | 0.486 |
| | avg | 0.486 | 0.443 | 0.542 | 0.496 | 0.512 | 0.466 | 0.498 | 0.455 | 0.494 | 0.455 | 0.744 | 0.602 | 0.512 | 0.465 | 0.496 | 0.451 |
| CzeLan | 96 | 0.178 | 0.229 | 0.231 | 0.310 | 0.176 | 0.237 | 0.185 | 0.253 | 0.180 | 0.232 | 0.184 | 0.262 | 0.183 | 0.251 | 0.211 | 0.289 |
| | 192 | 0.210 | 0.252 | 0.268 | 0.337 | 0.215 | 0.279 | 0.214 | 0.286 | 0.214 | 0.258 | 0.232 | 0.300 | 0.208 | 0.271 | 0.252 | 0.323 |
| | 336 | 0.243 | 0.280 | 0.298 | 0.361 | 0.224 | 0.288 | 0.248 | 0.311 | 0.248 | 0.289 | 0.287 | 0.357 | 0.243 | 0.302 | 0.317 | 0.366 |
| | 720 | 0.276 | 0.317 | 0.410 | 0.401 | 0.282 | 0.337 | 0.279 | 0.339 | 0.278 | 0.329 | 0.405 | 0.445 | 0.273 | 0.335 | 0.358 | 0.392 |
| | avg | 0.227 | 0.270 | 0.302 | 0.352 | 0.224 | 0.285 | 0.232 | 0.297 | 0.230 | 0.277 | 0.277 | 0.341 | 0.227 | 0.290 | 0.285 | 0.343 |
| COVID-19 | 24 | 4.458 | 1.230 | 4.869 | 1.394 | 4.799 | 1.347 | 4.715 | 1.321 | 6.335 | 1.554 | 5.643 | 1.424 | 5.528 | 1.450 | 9.780 | 1.851 |
| | 36 | 6.842 | 1.624 | 7.376 | 1.708 | 7.536 | 1.727 | 7.299 | 1.681 | 8.222 | 1.787 | 9.141 | 1.848 | 8.351 | 1.830 | 12.804 | 2.083 |
| | 48 | 10.213 | 2.009 | 10.051 | 1.999 | 10.131 | 2.130 | 10.141 | 2.012 | 11.669 | 2.157 | 10.904 | 2.303 | 11.259 | 2.114 | 14.244 | 2.189 |
| | 60 | 12.237 | 2.198 | 11.764 | 2.119 | 12.582 | 2.272 | 11.871 | 2.156 | 12.188 | 2.173 | 12.688 | 2.168 | 12.666 | 2.225 | 15.472 | 2.275 |
| | avg | 8.438 | 1.765 | 8.515 | 1.805 | 8.762 | 1.808 | 8.506 | 1.793 | 9.604 | 1.918 | 9.594 | 1.936 | 9.451 | 1.905 | 13.075 | 2.099 |
| Wiki | 24 | 6.532 | 0.440 | 6.811 | 0.464 | 6.624 | 0.432 | 6.886 | 0.437 | 6.900 | 0.446 | 7.023 | 0.512 | 6.858 | 0.430 | 6.883 | 0.520 |
| | 36 | 5.948 | 0.442 | 6.341 | 0.479 | 6.038 | 0.453 | 6.431 | 0.452 | 6.520 | 0.467 | 6.922 | 0.495 | 6.400 | 0.445 | 6.393 | 0.538 |
| | 48 | 5.784 | 0.459 | 5.895 | 0.509 | 5.874 | 0.464 | 6.101 | 0.483 | 6.108 | 0.484 | 6.841 | 0.514 | 5.959 | 0.449 | 5.940 | 0.547 |
| | 60 | 5.489 | 0.462 | 5.546 | 0.514 | 5.493 | 0.463 | 5.681 | 0.466 | 5.732 | 0.476 | 5.850 | 0.546 | 5.633 | 0.452 | 5.605 | 0.552 |
| | avg | 5.938 | 0.451 | 6.148 | 0.492 | 6.007 | 0.453 | 6.275 | 0.460 | 6.315 | 0.468 | 6.659 | 0.517 | 6.212 | 0.444 | 6.205 | 0.539 |
| 1st Count | | 18 | 19 | 2 | 2 | 3 | 0 | 0 | 0 | 0 | 0 | 0 | 0 | 2 | 4 | 0 | 0 |

**Performance on Extra Benchmarks.** We evaluate the performance of Twinsformer on more new real-world datasets from fev-bench (Shchur et al., 2025) or GIFT-Eval (Aksu et al., 2024):

- Wind (Li et al., 2022) provides predicted wind speeds for a specific location, with a temporal resolution of 15 minutes. Each data point represents a one-hour-ahead forecast, and the dataset covers the period from January 1 to February 1, 2020.

- ZafNoo (Qiu et al., 2024) comprises solar irradiance measurements with a half-hourly temporal resolution, covering the period between mid-May and late June of 2008.

- CzeLan (Qiu et al., 2024) contains time-series monitoring data from the Czech Republic (CZE), recorded at a consistent 30-minute interval between May and June 2016.

- COVID-19 (Chen et al., 2022), provided by Johns Hopkins University, maintains daily records of COVID-19 hospitalizations in California from February to December 2020.

- Wiki [2] contains daily page views for 60,000 Wikipedia articles in eight languages over 2018-2019, where we subsequently selected the first 99 articles as our experimental subset.

To ensure fair comparisons, we applied a fixed chronological split ratio of either 7:1:2 or 6:2:2 for training, validation, and testing across all datasets. However, due to the limited total length of the Wiki and COVID-19 datasets, we adopted a specific forecasting setting with an input length of 36 and prediction lengths of $\{24, 36, 48, 60\}$. For the other, longer datasets, we used an input length of 96 and forecasting lengths of $\{96, 192, 336, 720\}$. As shown in Table 9, the comprehensive results indicate that our TwinsFormer consistently outperforms state-of-the-art models.

**Comparison with DESTformer.** DESTformer (Wang et al., 2023b) is a trend-seasonal decomposition-based Transformer framework, which leverages a multi-scale attention and a multi-view attention mechanism to capture fine-grained temporal patterns. In Table 10, we compare the

---

[2] https://www.kaggle.com/datasets/sandeshbhat/wikipedia-webtraffic-201819.

Table 10: Comparison with DESTformer on three datasets.

| Datasets | | Weather | | | | | ECL | | | | | Traffic | | | | |
|---|---|---|---|---|---|---|---|---|---|---|---|---|---|---|---|---|
| Metrics | | 96 | 192 | 336 | 720 | Avg | 96 | 192 | 336 | 720 | Avg | 96 | 192 | 336 | 720 | Avg |
| **TwinsFormer** | MSE | **0.158** | **0.207** | **0.263** | **0.339** | **0.242** | **0.134** | **0.154** | **0.165** | **0.198** | **0.163** | **0.379** | **0.388** | **0.407** | **0.439** | **0.403** |
| **(Ours)** | MAE | **0.199** | **0.243** | **0.285** | **0.336** | **0.266** | **0.223** | **0.240** | **0.257** | **0.290** | **0.253** | **0.258** | **0.265** | **0.272** | **0.289** | **0.271** |
| DESTformer | MSE | 0.184 | 0.237 | 0.289 | 0.365 | 0.269 | 0.165 | 0.176 | 0.188 | 0.219 | 0.187 | 0.427 | 0.453 | 0.468 | 0.488 | 0.459 |
| (Reproduced) | MAE | 0.225 | 0.264 | 0.316 | 0.379 | 0.296 | 0.258 | 0.263 | 0.274 | 0.313 | 0.277 | 0.274 | 0.298 | 0.315 | 0.323 | 0.303 |
| DESTformer | MSE | 0.202 | 0.254 | 0.307 | 0.395 | 0.290 | 0.187 | 0.195 | 0.203 | 0.229 | 0.204 | 0.562 | 0.593 | 0.603 | 0.599 | 0.589 |
| (Original) | MAE | 0.276 | 0.318 | 0.369 | 0.417 | 0.345 | 0.296 | 0.307 | 0.325 | 0.338 | 0.317 | 0.354 | 0.355 | 0.377 | 0.356 | 0.361 |

performance of TwinsFormer and DESTformer across Weather, ECL, and Traffic datasets. Since the author of DESTformer did not provide the open-source code repository, we reimplemented DEST-former based on the algorithm in the original paper and reproduced the results with the same training strategies as TwinsFormer. Meanwhile, we incorporated the original results of DESTformer as a reference. As can be observed in Table 10, although the reproduced results of DESTformer are much better than those in the original paper, the performance is still inferior to our method, which fully demonstrates the effectiveness and superiority of TwinsFormer. The following points are worth noting when interpreting the results:

- DESTformer designs two specialized, complex attention mechanisms to process trend and seasonal components separately, which overlooks the real-world complexity where seasonal and trend information are often entangled.

- The FFT-based decomposition in DESTformer is primarily a denoising operation, which might discard meaningful, non-periodic signals that are not captured by the simple trend component, potentially losing valuable information.

- TwinsFormer allows the model to utilize more information from the original signal. The "noise" is not just removed but is repurposed to refine the representations of both components, leading to a more accurate performance for time series forecasting.

Table 11: Zero-shot forecasting results on ETT datasets.

| Model | | TwinsFormer | | WPMixer | | Fredformer | | TimeMixer | | iTransformer | | PatchTST | |
|---|---|---|---|---|---|---|---|---|---|---|---|---|---|---|
| | | (Ours) | | 2025 | | 2024 | | 2024 | | 2024 | | 2023 | |
| Metric | | MSE | MAE | MSE | MAE | MSE | MAE | MSE | MAE | MSE | MAE | MSE | MAE |
| ETTh1⟶ETTh2 | 96 | **0.301** | **0.342** | 0.318 | 0.351 | 0.317 | 0.374 | 0.313 | 0.368 | 0.314 | 0.366 | 0.313 | 0.362 |
| | 192 | **0.375** | **0.394** | 0.382 | 0.402 | 0.392 | 0.406 | 0.401 | 0.406 | 0.392 | 0.417 | 0.396 | 0.412 |
| | 336 | **0.418** | **0.427** | 0.435 | 0.453 | 0.421 | 0.435 | 0.428 | 0.440 | 0.435 | 0.436 | 0.433 | 0.439 |
| | 720 | **0.425** | **0.438** | 0.454 | 0.472 | 0.436 | 0.449 | 0.439 | 0.464 | 0.444 | 0.457 | 0.442 | 0.453 |
| | Avg | **0.380** | **0.400** | 0.397 | 0.420 | 0.392 | 0.416 | 0.395 | 0.420 | 0.396 | 0.419 | 0.396 | 0.417 |
| ETTm1⟶ETTm2 | 96 | 0.184 | **0.264** | **0.178** | 0.285 | 0.213 | 0.270 | 0.192 | 0.267 | 0.186 | 0.268 | 0.195 | 0.271 |
| | 192 | 0.256 | **0.298** | **0.253** | 0.331 | 0.274 | 0.315 | 0.268 | 0.312 | 0.263 | 0.317 | 0.258 | 0.311 |
| | 336 | 0.313 | **0.336** | **0.307** | 0.374 | 0.326 | 0.363 | 0.317 | 0.347 | 0.311 | 0.352 | 0.317 | 0.348 |
| | 720 | **0.408** | **0.409** | 0.410 | 0.433 | 0.421 | 0.422 | 0.416 | 0.412 | 0.421 | 0.417 | 0.416 | 0.414 |
| | Avg | 0.290 | **0.327** | **0.287** | 0.356 | 0.309 | 0.343 | 0.298 | 0.335 | 0.295 | 0.339 | 0.297 | 0.336 |
| ETTh1⟶ETTm2 | 96 | 0.219 | **0.248** | 0.239 | 0.278 | 0.235 | 0.275 | **0.218** | 0.263 | 0.229 | 0.255 | 0.225 | 0.257 |
| | 192 | **0.264** | 0.327 | 0.286 | 0.327 | 0.283 | 0.323 | 0.265 | 0.343 | 0.274 | **0.321** | 0.268 | 0.323 |
| | 336 | 0.343 | 0.379 | 0.331 | 0.402 | **0.325** | 0.395 | 0.326 | 0.382 | 0.330 | **0.376** | 0.332 | 0.385 |
| | 720 | **0.429** | **0.445** | 0.474 | 0.501 | 0.464 | 0.473 | 0.442 | 0.494 | 0.453 | 0.486 | 0.447 | 0.481 |
| | Avg | 0.314 | **0.350** | 0.333 | 0.377 | 0.327 | 0.367 | **0.313** | 0.371 | 0.322 | 0.360 | 0.318 | 0.362 |
| ETTm1⟶ETTh2 | 96 | **0.353** | **0.368** | 0.376 | 0.384 | 0.362 | 0.388 | 0.364 | 0.377 | 0.359 | 0.380 | 0.381 | 0.389 |
| | 192 | **0.392** | **0.409** | 0.413 | 0.421 | 0.415 | 0.429 | 0.411 | 0.412 | 0.408 | 0.419 | 0.419 | 0.423 |
| | 336 | **0.467** | **0.472** | 0.478 | 0.496 | 0.473 | 0.511 | 0.473 | 0.479 | 0.455 | 0.482 | 0.488 | 0.494 |
| | 720 | **0.504** | **0.521** | 0.544 | 0.558 | 0.539 | 0.559 | 0.522 | 0.543 | 0.531 | 0.545 | 0.554 | 0.569 |
| | Avg | **0.429** | **0.443** | 0.453 | 0.465 | 0.447 | 0.472 | 0.443 | 0.453 | 0.438 | 0.457 | 0.461 | 0.469 |

**Extra Generalization and Robustness Analysis.** To further evaluate the generalization and robustness ability of TwinsFormer, we conduct extensive experiments under zero-shot, noisy, and missing data settings. For the zero-short setting, we utilize the models trained on one dataset to evaluate on

another without retraining directly. Furthermore, we evaluate model robustness under covariate shift by corrupting testing inputs with additive noise and random missing values. Specifically, we inject white Gaussian noise into the test series $X_{test}$ to generate a corrupted version:

$$X_{corrupted} = X_{test} + \epsilon, \quad \epsilon \sim \mathcal{N}(0, \sigma^2), \quad \sigma = 0.1 \times \sigma_x, \tag{26}$$

where $\sigma_x$ is the standard deviation of the training set, ensuring the noise level is scaled appropriately for each dataset. Meanwhile, we randomly set values in the testing sequence to *NaN* with a probability of $p = 10\%$. A simple forward-fill imputation is applied to maintain the input dimensions. Crucially, all models are evaluated on the corrupted test sets without any retraining or fine-tuning, testing their inherent robustness to imperfect data.

As shown in Table 11, our method notably outperforms other models, which indicates the superority of TwinsFormer in the cross-domain learning capability. Moreover, TwinsFormer exhibits smaller performance degradation compared to baselines in Table 12, suggesting our interactive design can effectively filter out noise and recover from localized missingness when the fundamental trend-seasonality decomposition remains valid.

Table 12: Robustness analysis with noise and missing data on ECL and Traffic datasets.

| Setups | | ECL | | | | | | Traffic | | | | | |
|---|---|---|---|---|---|---|---|---|---|---|---|---|---|
| | | Clean | | Noise ($\sigma = 0.1$) | | Missing ($p = 10\%$) | | Clean | | Noise ($\sigma = 0.1$) | | Missing ($p = 10\%$) | |
| Metric | | MSE | MAE | MSE | MAE | MSE | MAE | MSE | MAE | MSE | MAE | MSE | MAE |
| Fredformer | 96 | 0.147 | 0.241 | 0.157 | 0.253 | 0.162 | 0.260 | 0.406 | 0.277 | 0.435 | 0.292 | 0.447 | 0.302 |
| | 192 | 0.165 | 0.258 | 0.176 | 0.271 | 0.181 | 0.278 | 0.426 | 0.290 | 0.457 | 0.306 | 0.469 | 0.316 |
| | 336 | 0.177 | 0.273 | 0.189 | 0.287 | 0.195 | 0.295 | 0.437 | 0.292 | 0.469 | 0.309 | 0.482 | 0.319 |
| | 720 | 0.213 | 0.304 | 0.227 | 0.319 | 0.234 | 0.328 | 0.462 | 0.305 | 0.496 | 0.323 | 0.509 | 0.334 |
| | Avg | 0.176 | 0.269 | 0.187 | 0.283 | 0.193 | 0.290 | 0.433 | 0.291 | 0.464 | 0.308 | 0.477 | 0.318 |
| iTransformer | 96 | 0.148 | 0.240 | 0.159 | 0.254 | 0.164 | 0.261 | 0.395 | 0.268 | 0.427 | 0.285 | 0.440 | 0.295 |
| | 192 | 0.162 | 0.253 | 0.174 | 0.268 | 0.179 | 0.275 | 0.417 | 0.276 | 0.451 | 0.294 | 0.464 | 0.304 |
| | 336 | 0.178 | 0.269 | 0.191 | 0.285 | 0.197 | 0.293 | 0.433 | 0.283 | 0.468 | 0.302 | 0.482 | 0.312 |
| | 720 | 0.225 | 0.317 | 0.241 | 0.334 | 0.248 | 0.343 | 0.467 | 0.302 | 0.505 | 0.323 | 0.520 | 0.334 |
| | Avg | 0.178 | 0.270 | 0.191 | 0.285 | 0.197 | 0.293 | 0.428 | 0.282 | 0.463 | 0.301 | 0.477 | 0.311 |
| TimeMixer | 96 | 0.153 | 0.244 | 0.164 | 0.258 | 0.169 | 0.265 | 0.473 | 0.287 | 0.510 | 0.306 | 0.525 | 0.316 |
| | 192 | 0.168 | 0.259 | 0.180 | 0.274 | 0.186 | 0.282 | 0.486 | 0.294 | 0.525 | 0.314 | 0.540 | 0.324 |
| | 336 | 0.185 | 0.275 | 0.198 | 0.291 | 0.204 | 0.299 | 0.488 | 0.298 | 0.527 | 0.318 | 0.543 | 0.328 |
| | 720 | 0.227 | 0.312 | 0.243 | 0.330 | 0.250 | 0.339 | 0.536 | 0.314 | 0.579 | 0.336 | 0.596 | 0.347 |
| | Avg | 0.183 | 0.272 | 0.196 | 0.288 | 0.202 | 0.296 | 0.496 | 0.298 | 0.535 | 0.319 | 0.551 | 0.329 |
| TwinsFormer | 96 | 0.134 | 0.223 | **0.141** | **0.232** | **0.145** | **0.238** | 0.379 | 0.258 | **0.402** | **0.270** | **0.411** | **0.278** |
| | 192 | 0.154 | 0.240 | **0.162** | **0.250** | **0.166** | **0.256** | 0.388 | 0.265 | **0.412** | **0.278** | **0.421** | **0.286** |
| | 336 | 0.165 | 0.257 | **0.173** | **0.267** | **0.178** | **0.274** | 0.407 | 0.272 | **0.432** | **0.286** | **0.442** | **0.294** |
| | 720 | 0.198 | 0.290 | **0.208** | **0.302** | **0.214** | **0.310** | 0.439 | 0.289 | **0.466** | **0.304** | **0.477** | **0.313** |
| | Avg | 0.163 | 0.253 | **0.171** | **0.263** | **0.176** | **0.270** | 0.403 | 0.271 | **0.428** | **0.285** | **0.438** | **0.293** |

**Clarification on Moving Average Decomposition.** We use a fixed kernel size of $k = 25$ for the moving average pooling, which is consistent across all datasets and settings. This value was chosen based on empirical validation and aligns with common practices in time series decomposition (e.g., Autoformer, FEDformer). To prevent data leakage and preserve temporal alignment, we perform padding using only values from within the historical input window, where the front padding and back padding are applied with the first and last values of the inputs, respectively. In abrupt trend changes or irregular sampling cases, the residual component $R$ (as shown in Figure 1) captures these irregularities, and our interactive module is designed to adaptively reassign such information between the trend and seasonal branches.

Actually, the moving average operation serves primarily as an initialization to provide a preliminary separation of trend and seasonal components. This is a starting point, not the final decomposition:

- The complex, non-stationary nature of real-world time series means that a fixed, linear decomposition is indeed insufficient. Our key innovation lies in the subsequent interactive modules, which enable the model to dynamically refine and recalibrate these initial components throughout the Transformer blocks.
- In essence, the moving average provides a strong inductive bias. TwinsFormer then learns to decompose more effectively by allowing the trend and seasonal branches to interact

and exchange information, progressively disentangling the two patterns in a data-driven manner. This process is far more powerful than a single, static decomposition.

Table 13: Ablation study on the kernel size $k$ of the moving-average initialization.

| TwinsFormer | | ECL | | Traffic | | Weather | | Solar | | ETTm1 | | ETTm2 | |
|---|---|---|---|---|---|---|---|---|---|---|---|---|---|
| Kernel Size | Metric | MSE | MAE | MSE | MAE | MSE | MAE | MSE | MAE | MSE | MAE | MSE | MAE |
| | 96 | 0.137 | 0.226 | 0.382 | 0.262 | 0.160 | 0.200 | 0.189 | 0.224 | 0.319 | 0.355 | 0.170 | 0.252 |
| | 192 | 0.163 | 0.243 | 0.396 | 0.270 | 0.210 | 0.244 | 0.220 | 0.245 | 0.365 | 0.385 | 0.237 | 0.290 |
| 5 | 336 | 0.172 | 0.265 | 0.410 | 0.276 | 0.268 | 0.286 | 0.242 | 0.268 | 0.397 | 0.403 | 0.293 | 0.334 |
| | 720 | 0.203 | 0.297 | 0.441 | 0.294 | 0.342 | 0.337 | 0.240 | 0.270 | 0.458 | 0.440 | 0.400 | 0.398 |
| | Avg | **0.169** | **0.258** | **0.407** | **0.276** | **0.245** | **0.267** | **0.223** | **0.252** | **0.385** | **0.396** | **0.275** | **0.319** |
| | 96 | 0.134 | 0.223 | 0.380 | 0.258 | 0.158 | 0.199 | 0.188 | 0.222 | 0.314 | 0.353 | 0.168 | 0.250 |
| | 192 | 0.154 | 0.240 | 0.389 | 0.266 | 0.209 | 0.244 | 0.219 | 0.247 | 0.361 | 0.383 | 0.235 | 0.288 |
| 15 | 336 | 0.166 | 0.258 | 0.408 | 0.273 | 0.265 | 0.286 | 0.240 | 0.265 | 0.395 | 0.401 | 0.291 | 0.329 |
| | 720 | 0.200 | 0.291 | 0.442 | 0.291 | 0.338 | 0.338 | 0.239 | 0.269 | 0.456 | 0.438 | 0.396 | 0.396 |
| | Avg | **0.164** | **0.253** | **0.405** | **0.272** | **0.241** | **0.266** | **0.222** | **0.251** | **0.382** | **0.394** | **0.273** | **0.316** |
| | 96 | 0.134 | 0.223 | 0.379 | 0.258 | 0.158 | 0.199 | 0.188 | 0.222 | 0.315 | 0.354 | 0.169 | 0.251 |
| | 192 | 0.154 | 0.240 | 0.388 | 0.265 | 0.207 | 0.243 | 0.219 | 0.246 | 0.362 | 0.384 | 0.236 | 0.289 |
| 25 | 336 | 0.165 | 0.257 | 0.407 | 0.272 | 0.263 | 0.285 | 0.240 | 0.265 | 0.396 | 0.402 | 0.292 | 0.330 |
| | 720 | 0.198 | 0.290 | 0.439 | 0.289 | 0.339 | 0.336 | 0.236 | 0.269 | 0.457 | 0.439 | 0.397 | 0.397 |
| | Avg | **0.163** | **0.253** | **0.403** | **0.271** | **0.242** | **0.266** | **0.221** | **0.251** | **0.383** | **0.395** | **0.274** | **0.317** |
| | 96 | 0.133 | 0.222 | 0.380 | 0.259 | 0.159 | 0.200 | 0.189 | 0.223 | 0.316 | 0.355 | 0.170 | 0.252 |
| | 192 | 0.158 | 0.244 | 0.389 | 0.266 | 0.208 | 0.245 | 0.220 | 0.245 | 0.363 | 0.385 | 0.237 | 0.290 |
| 35 | 336 | 0.168 | 0.263 | 0.408 | 0.273 | 0.265 | 0.286 | 0.242 | 0.264 | 0.397 | 0.403 | 0.293 | 0.331 |
| | 720 | 0.203 | 0.292 | 0.441 | 0.290 | 0.342 | 0.338 | 0.237 | 0.270 | 0.458 | 0.440 | 0.398 | 0.398 |
| | Avg | **0.166** | **0.255** | **0.405** | **0.272** | **0.243** | **0.267** | **0.222** | **0.251** | **0.384** | **0.396** | **0.274** | **0.318** |
| | 96 | 0.138 | 0.229 | 0.387 | 0.266 | 0.161 | 0.203 | 0.192 | 0.225 | 0.320 | 0.357 | 0.172 | 0.254 |
| | 192 | 0.162 | 0.248 | 0.396 | 0.273 | 0.212 | 0.248 | 0.223 | 0.252 | 0.367 | 0.388 | 0.238 | 0.294 |
| 45 | 336 | 0.170 | 0.265 | 0.415 | 0.280 | 0.271 | 0.293 | 0.241 | 0.267 | 0.401 | 0.402 | 0.294 | 0.336 |
| | 720 | 0.208 | 0.298 | 0.447 | 0.297 | 0.347 | 0.344 | 0.243 | 0.271 | 0.458 | 0.442 | 0.401 | 0.400 |
| | Avg | **0.170** | **0.260** | **0.411** | **0.279** | **0.248** | **0.272** | **0.225** | **0.254** | **0.387** | **0.397** | **0.276** | **0.321** |

To directly address the concern about the reliance on a specific initialization, we conducted a systematic ablation on the kernel size $k$. The results, presented in Table 13, show that TwinsFormer's performance remains stable and superior across a wide range of kernel sizes, from a very local context ($k = 13$) to a much smoother one ($k = 45$). The negligible performance variance (Avg MSE range of 0.002) clearly indicates that:

- The interactive modules in TwinsBlocks are the primary drivers of performance, as they can effectively refine a wide range of initial decompositions.
- Our method is highly robust and generalizable, as it does not require careful tuning of the decomposition kernel for different datasets.

**Justification of Interactive Module Design.** The interaction module acts as a "correction mechanism". The seasonal branch, being processed by the powerful Attention and FFN modules, learns rich representations of dependencies and fine-grained temporal variations. The four MLPs are trained end-to-end with the entire model via backpropagation, which transforms these signals into "guidance" (scaling and shifting factors) for the trend branch. This allows the trend embedding to be adaptively updated, absorbing or discarding information that was initially mis-assigned by the simple moving average. To rigorously validate this design choice, we conducted a fine-grained ablation study on Table 14. We systematically compared our full model against several variants:

- **Variant A**: Only using learnable bias MLPs ($\beta, \mu$), and scaling MLPS ($\alpha, \gamma$) are fixed to 1.
- **Variant B**: Only using learnable scaling MLPS ($\alpha, \gamma$), and bias MLPs ($\beta, \mu$) are fixed to 0.
- **Variant C**: $A_S$ and $F_S$ share the same MLPs, i.e. $\alpha = \gamma, \beta = \mu$.

- **Variant D**: Only using $A_S$ for interaction and ignoring $F_S$.
- **Variant E**: Replacing four MLPs with a single FiLM-layer (Perez et al., 2018) conditioned only on $A_S$.
- **Variant F**: Two separate FiLM-layers for $A_S$ and $F_S$.

Table 14: Fine-grained ablation study on our interactive module.

| Variant | ETTm1 | | ETTh2 | | ECL | | Traffic | | Weather | | Solar | | PEMS03 | | PEMS07 | |
|---|---|---|---|---|---|---|---|---|---|---|---|---|---|---|---|---|
| | MSE | MAE | MSE | MAE | MSE | MAE | MSE | MAE | MSE | MAE | MSE | MAE | MSE | MAE | MSE | MAE |
| TwinsFormer | **0.315** | **0.354** | **0.285** | **0.332** | **0.134** | **0.223** | **0.379** | **0.258** | **0.158** | **0.199** | **0.188** | **0.222** | **0.063** | **0.165** | **0.055** | **0.145** |
| A | 0.327 | 0.368 | 0.296 | 0.345 | 0.141 | 0.231 | 0.395 | 0.272 | 0.165 | 0.207 | 0.201 | 0.234 | 0.070 | 0.170 | 0.062 | 0.152 |
| B | 0.321 | 0.359 | 0.290 | 0.339 | 0.139 | 0.228 | 0.388 | 0.264 | 0.163 | 0.205 | 0.194 | 0.229 | 0.079 | 0.179 | 0.060 | 0.149 |
| C | 0.320 | 0.360 | 0.288 | 0.340 | 0.137 | 0.226 | 0.384 | 0.261 | 0.161 | 0.202 | 0.192 | 0.225 | 0.068 | 0.168 | 0.058 | 0.147 |
| D | 0.334 | 0.375 | 0.309 | 0.353 | 0.148 | 0.242 | 0.406 | 0.281 | 0.172 | 0.215 | 0.215 | 0.242 | 0.083 | 0.184 | 0.074 | 0.166 |
| E | 0.329 | 0.372 | 0.302 | 0.349 | 0.143 | 0.235 | 0.398 | 0.275 | 0.166 | 0.208 | 0.206 | 0.236 | 0.078 | 0.177 | 0.070 | 0.160 |
| F | 0.324 | 0.363 | 0.299 | 0.347 | 0.142 | 0.233 | 0.389 | 0.265 | 0.164 | 0.206 | 0.200 | 0.232 | 0.075 | 0.172 | 0.068 | 0.155 |

As shown in Table 14, our interactive module design is optimal, and removing components (e.g., scaling or bias terms) in Variants A, B, C, and D leads to performance degradation, demonstrating that our four-MLP design with dedicated processing pathways provides significantly better representation capacity for handling complex temporal interactions. Crucially, even the FiLM-conditioning Variants E and F cannot match our performance, validating our architectural innovation in using specialized MLPs for distinct signal types. Although both FiLM and our interactive module use affine transformations, our interactive module represents a significant architectural innovation specifically designed for time series decomposition:

- **Novel Dual-Source Conditioning**: We process two distinct signals ($A_S$ for cross-variate dependencies and $F_S$ for temporal details) through separate pathways, unlike single-input-based FiLM.
- **Specialized Transformation Networks**: We use dedicated MLPs for each type of signal, enabling specialized processing of different information characteristics.
- **Decomposition-Specific Design**: Our module is integral to a novel dual-stream architecture for component interaction, representing a new application domain for feature-wise modulation.

**Performance on the Irregular Multivariate Time Series Forecasting.** To evaluate the performance on irregularly sampled time series, we use four datasets and follow the setting of (Zhang et al., 2024a). Here is the detailed information for these datasets:

- Human Activity [3] consists of 12 irregularly measured 3D positional variables from sensors worn on the ankles, belts, and chests of five individuals performing various activities.
- USHCN [4] includes over 150 years of climate data from multiple U.S. stations, covering 5 climate variables.
- PhysioNet [5] includes 12000 IMTS from different patients, each with 41 clinical signals collected irregularly during the first 48 hours of ICU admission.
- MIMIC-III [6] is a widely accessible clinical database that houses electronic health records of patients in critical care.

As shown in Table 15, the results indicate that Twinsformer is inferior to some baselines. The performance drop on irregularly-sampled data stems from a fundamental architectural premise: both the moving average and Fourier decomposition modules necessitate a regular time grid to produce well-defined trend and seasonal components. The subsequent interactive mechanism, designed to

---

[3] https://archive.ics.uci.edu/dataset/196/localization+data+for+person+activity

[4] https://www.osti.gov/biblio/1394920

[5] https://archive.physionet.org/challenge/2012

[6] https://mimic.mit.edu/

refine these semantically meaningful representations, cannot recover from the inherently flawed inputs generated under irregular sampling, causing the model's core advantage to diminish.

Table 15: Performance comparison for irregular multivariate time series forecasting.

| Setups | Human Activity | | USHCN | | PhysioNet | | MIMIC-III | |
|---|---|---|---|---|---|---|---|---|
| | 2000ms⟶2000ms | | 24months⟶6months | | 36h⟶12h | | 36h⟶12h | |
| Metric | MSE | MAE | MSE | MAE | MSE | MAE | MSE | MAE |
| PatchTST | 0.008 | 0.064 | 0.615 | 0.409 | 0.028 | 0.115 | 0.098 | 0.243 |
| iTransformer | 0.012 | 0.082 | 0.608 | 0.398 | 0.065 | 0.214 | 0.073 | 0.216 |
| TimeMixer | 0.006 | 0.054 | 0.596 | 0.370 | 0.013 | 0.129 | 0.046 | 0.137 |
| Fredformer | 0.007 | 0.064 | 0.605 | 0.389 | **0.011** | **0.060** | **0.022** | **0.088** |
| WPMixer | **0.005** | **0.050** | **0.568** | **0.368** | 0.028 | 0.072 | 0.057 | 0.117 |
| TwinsFormer | 0.006 | 0.066 | 0.602 | 0.385 | 0.032 | 0.084 | 0.078 | 0.210 |

**Extra Visualization for the Interaction Mechanism.** To better understand our interactive module, we compare the intrinsic representations among: (1) raw time series, (2) moving average decomposition, and (3) the enhanced output of our interactive module. Specifically, we employ the t-SNE (Maaten & Hinton, 2008) tool to qualitatively compare the intrinsic structure of different feature representations. While t-SNE provides nonlinear projections that may distort absolute distances, the consistent experimental setup across all three conditions allows for meaningful relative comparisons of representation quality and temporal coherence.

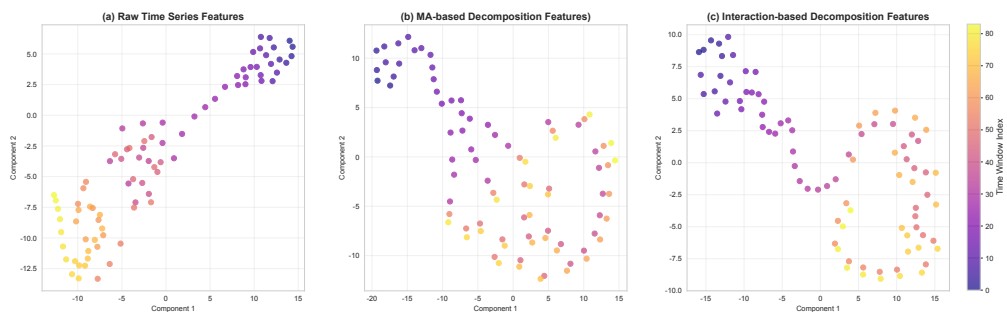

Figure 9: Visualization of the learned features using t-SNE on ETTh2.

As shown in Figure 9, the raw time series features exhibit a scattered distribution with intermixed points from different time windows, indicating ambiguous intrinsic structure and poorly defined temporal patterns. In comparison, the moving average (MA)-based decomposition features demonstrate improved clustering over the raw representation, suggesting that MA decomposition can partially extract meaningful temporal structures. Notably, the interaction-based decomposition features reveal significantly clearer and more organized clustering patterns. The coherent progression of points corresponding to sequential time windows provides visual evidence that our interactive module effectively captures intrinsic temporal dependencies and produces more structured representations.

**Limitations and Future Works.** Despite its compelling performance on regularly-sampled time series, our work has a clear boundary for the current model's applicability: it is highly effective for forecasting in domains with strong periodicity and regular sampling (e.g., energy, traffic, weather) but is not yet suited for inherently irregular time series. Future work will be directed toward transcending this boundary to create a more universal forecasting framework. We plan to explore continuous decomposition strategies with neural ordinary differential equations or continuous-time state-space models, which can inherently model the latent trend and seasonal dynamics directly from irregular observations and provide a robust foundation for the dual streams. Furthermore, although the interaction module is designed to be lightweight, the computational complexity inherent in the attention mechanism remains prohibitive for high-dimensional variable datasets. Thus, the development of an interaction framework based on non-attention mechanisms represents a promising direction for future research.

