# OpenReview forum: "TwinsFormer: Revisiting Inherent Dependencies via Two Interactive Components for Time Series Forecasting"
_ICLR.cc/2026/Conference — Submitted to ICLR 2026_

### Official Review · Reviewer_ferD · 2025-10-25

**Soundness:** 3
**Presentation:** 3
**Contribution:** 3
**Rating:** 6
**Confidence:** 3

**Summary:**

This work presents a new Transformer-based architecture that innovatively integrates time series decomposition with interactive learning. The proposed dual-stream design and interactive mechanism represent an advancement in modeling temporal dependencies.

**Strengths:**

1. The dual-stream design with explicit interaction between trend and seasonal components is novel and well-executed. The interactive module that enables information flow between decomposition branches addresses a genuine limitation in existing decomposition-based methods.

2. The generalization bound analysis provides valuable theoretical insights into why the decomposition-interaction mechanism leads to improved performance, elevating the work beyond purely empirical contributions.

3. The thorough ablation experiments in Table 3 effectively validate the importance of each component, particularly demonstrating the value of the subtraction mechanism and interactive module over simpler alternatives.

**Weaknesses:**

1. While the paper demonstrates that the interactive module improves performance, it provides limited insight into what specific information is being exchanged between the trend and seasonal branches. A more detailed analysis of the interaction dynamics would strengthen the contribution.

2. The work relies on a simple moving average for decomposition initialization. The paper will benefit from exploring how sensitive the model is to different decomposition techniques and whether more sophisticated decomposition methods could yield further improvements.

**Questions:**

1. Can you provide more insight into what types of information are typically transferred between the seasonal and trend branches through your interactive module? Are there specific temporal patterns that trigger stronger interactions?

2. Have you experimented with more advanced decomposition techniques beyond moving average? Does the performance gain primarily come from the interaction mechanism or could it be limited by the decomposition quality?

---

> ### Author Response · Authors · 2025-11-21
>
> We sincerely thank Reviewer ferD for their thoughtful and constructive feedback. We are pleased to provide a point-by-point response below, which we believe further strengthens our contribution.
>
> W1&Q1: We thank the reviewer for this excellent suggestion. The primary function of the interactive module is to act as a $\textbf{correction mechanism}$, allowing the trend branch to be refined using signals identified by the more powerful seasonal branch (which is processed by Attention and FFN). Based on our design and analysis, we can provide deeper insight into the specific information being exchanged:
> * $\textbf{From Seasonal to Trend}$ ($A_S$ and $F_S$): The signals $A_S$ (attention-weighted seasonal representations) and $F_S$ (discarded signals from the seasonal FFN) are not merely raw data. $A_S$ encapsulates learned $\textbf{cross-variate dependencies}$, effectively telling the trend branch "which variates' seasonal fluctuations are most relevant for contextualizing the overall long-term direction." Concurrently, $F_S$ provides fine-grained, $\textbf{temporally local variations}$ that were filtered out of the seasonal component, suggesting "these high-frequency details might be more characteristic of a local trend adjustment than a global seasonal pattern."
> * $\textbf{What Triggers Interaction}$: The transformation of $A_S$ and $F_S$ into scaling ($\exp(α(A_S))$), $\exp(γ(F_S))$) and shifting ($β(A_S), μ(F_S)$) factors via MLPs allows the model to learn this dynamically. $\textbf{Stronger interactions}$ are triggered when there is a significant discrepancy between the simple moving-average trend and the complex patterns discovered by the seasonal branch. This is often the case with sudden $\textbf{trend shifts}$, $\textbf{anomalous events}$, or $\textbf{complex}$, $\textbf{non-stationary patterns}$ where the initial decomposition is imperfect. Our visualization in $\textbf{Figure 9}$ of the revised version provides empirical support, showing that the interactive module leads to more coherent and disentangled feature clusters than the initial moving average, demonstrating a successful transfer and assimilation of this information.
>
> W2&Q2: We thank the reviewer for this insightful question. This exact concern was a central part of our $\textbf{original experimental design}$. To preemptively address the role of decomposition quality, we systematically evaluated TwinsFormer's robustness using different decomposition initializations, with results presented in our original $\textbf{Table 4}$.
>
> The results in Table 4 demonstrate that TwinsFormer's performance is highly $\textbf{robust and not sensitive}$ to the specific choice of decomposition technique. This robust insensitivity allows us to definitively state that the performance gains are primarily attributable to our novel interaction mechanism, not to the quality of the initial decomposition.
>
> We thank the reviewer again for their valuable comments, and hope our responses and additional analyses adequately address the concerns raised.

---

> > ### Comment · Reviewer_ferD · 2025-11-22
> >
> > Thank you for the authors’ efforts in addressing my concerns. The responses have clarified my questions, and I have decided to raise my score from 6 to 8.

---

> > > ### Author Response · Authors · 2025-11-22
> > >
> > > Thank you for your positive feedback and for raising your score. We are delighted that our responses addressed your concerns. Your insights have been invaluable in improving our manuscript.

---

### Official Review · Reviewer_EqxD · 2025-10-28

**Soundness:** 2
**Presentation:** 3
**Contribution:** 2
**Rating:** 4
**Confidence:** 5

**Summary:**

TwinsFormer, a Transformer-based architecture for time series forecasting that specifically takes into account interactions between seasonal and decomposed trend components, is presented in this paper.  TwinsFormer creates interactive modules where trend and seasonal representations are entwined via attention and feed-forward mechanisms, such as gating and subtraction/complementary transformations, in contrast to current dual-branch approaches that function independently on decomposed parts.  Empirical results show that the model performs at the state-of-the-art level on most tasks when tested against 13 real-world benchmarks for both short-term and long-term forecasting.  Visual diagnostics such as component effect figures and results tables are presented along with ablation studies, compatibility analysis, and theoretical justifications (including generalization bounds).

**Strengths:**

1. Motivated Issue and Explicit Restrictions of Previous Research:  The authors critically analyze a common flaw in Transformers' time series decomposition, specifically the underutilization of the interaction between trend and seasonal components (see Figure 1 for a visualization and Section 1 for an explanation).  This solves a practical gap in the field.

2. Strong Empirical Assessment:  To compare against a wide range of extremely competitive baselines, including Transformer-based and linear/TCN approaches, extensive experiments are carried out on a variety of benchmarks (see Table 1 for long-term forecasting and Table 2 for short-term forecasting).  TwinsFormer consistently outperforms or performs on par with the best models, according to the results, particularly on difficult multivariate settings.  Several datasets and prediction lengths are examined.

3. Clarity of Presentation: Both theorists and practitioners can understand the work thanks to its combination of mathematical explanation, architectural diagrams (Figure 2), component visualizations, performance curves (Figure 4), and hyperparameter sensitivity plots (Figure 7).  Every visual component is closely related to a central claim or conclusion.

**Weaknesses:**

1. The paper omits comparison with closely related recent work (e.g., DESTformer, Modeling Temporal Symmetry, Beyond Trend and Periodicity) that also adopt trend–seasonality decomposition and dual-component designs. Without a careful contrast of assumptions, architectures, and results, it is hard to judge whether the claimed contribution is genuinely novel or an incremental extension, which weakens originality and completeness.

2. Theoretical analysis (e.g., Rademacher complexity in Section 3.3) is plausible, but empirical support for real-world generalization is loosely connected. Beyond the MSE gap in Fig. 4, there is no systematic evaluation on out-of-distribution data or failure scenarios. Robustness studies under noise, missingness, and distribution shift, as well as transfer settings, are missing.

3. Methodology is under-specified. For the moving-average decomposition, the kernel window, padding effects, and failure modes (irregular sampling, abrupt trend changes) are not detailed, raising even data-leakage concerns. The Interaction Module lacks intuition and analysis for why the four MLPs (α,β,γ,μ) are chosen and how they are trained. Performance may depend heavily on such low-level choices, which calls robustness and generality into question.

4. The “Limitations” section is superficial. It does not systematically address conditions where the dual-stream strategy can be suboptimal, such as weak seasonality, corrupted observations, or richer multi-scale structures. Concrete failure cases, complexity–performance trade-offs, and plans to improve scalability and robustness should be articulated.

**Questions:**

1. Can the authors use the same evaluation protocol to directly compare TwinsFormer with DESTformer's dual-component framework on at least two or three key benchmarks?
2. How does the moving average kernel for decomposition perform on datasets with weak or non-existent trend/seasonality, or on data that is non-stationary or irregularly sampled?

---

> ### Author Response · Authors · 2025-11-21
>
> We sincerely thank Reviewer Y1LV for the insightful comments and constructive feedback. We are encouraged by the positive recognition of our work’s motivation, empirical evaluation, and clarity. Below, we provide a point-by-point response to the concerns raised.
>
> W1&Q1: Comparison with DESTformer.
>
> We appreciate the reviewer’s suggestion to compare with the decomposition-based  DESTformer. We have now included a direct comparison with DESTformer on Weather, ECL, and Traffic datasets.  TwinsFormer achieves better performance than DESTformer. These results are included in $\textbf{Table 10}$ of the revised manuscript.
>
> W2:  Limited Empirical Support for Generalization.
>
> Thanks for your valuable suggestions. We agree that empirical validation under distribution shifts is important. In the revised version, we have added a systematic generalization analysis under zero-shot, noisy, and missing data settings in $\textbf{Tables 11-12}$. These results indicate that our TwinsFormer has better generalization capability.
>
> W3: Under-Specified Methodology.
>
> We sincerely thank the reviewer for these critical points. We apologize for the lack of clarity and are pleased to provide the following elucidations:
> * The reviewer rightly points out the potential dependency on the decomposition method. We would like to clarify that the moving average kernel is only one of the possible decomposition initializers for our core interactive architecture. The novelty of TwinsFormer lies not in the decomposition itself, but in the interactive learning between the resulting components.
> In the original manuscript, we have already conducted experiments using two alternative decomposition methods in $\textbf{Table 4}$ for comparison. The results show that the performance gap between different decomposition methods is minimal, demonstrating that TwinsFormer's performance is robust to the specific choice of decomposition technique.
> * We have supplemented the justification for the moving average decomposition and interactive module in the revised manuscript. Specifically, we added ablations for the kernel size and the interactive module. Please see $\textbf{Tables 13-14}$ of the revised version for more details.
>
> W4&Q2: Expanded Limitations and Performance on Non-Stationary or Weakly Seasonal Data.
>
> We thank the reviewer for these insightful points. We have conducted experiments on irregular multivariate time series forecasting ($\textbf{Table 15}$) and added a subsection for $\textbf{Limitations and future works}$ in the revised manuscript.
>
> We thank the reviewer again for their valuable comments, and hope our responses and additional analyses adequately address the concerns raised.

---

> > ### Comment · Reviewer_EqxD · 2025-11-24
> >
> > Thank you very much for the thorough revisions and detailed clarifications. After reading the updated manuscript and your responses, I see that the concerns I initially raised have been sufficiently addressed.
> >
> > Overall, my questions have been resolved, and I am satisfied with the additional experiments and explanations provided. I thank the authors for their efforts in improving the manuscript.
> >
> > I will raise my score from 4 to 6.

---

> > > ### Author Response · Authors · 2025-11-24
> > >
> > > We would like to thank Reviewer EqxD for providing a detailed and valuable review, which helps us a lot in the rebuttal and paper revision.
> > > Thank you again for your response and for raising the score!

---

### Official Review · Reviewer_Y1LV · 2025-10-29

**Soundness:** 2
**Presentation:** 2
**Contribution:** 2
**Rating:** 4
**Confidence:** 5

**Summary:**

This paper designs two interactive components for information mining of time series data.

**Strengths:**

The solution proposed in this article is simple and easy to understand.

**Weaknesses:**

•  I understand that the residuals contain interactive and other useful components, but this may result from the decomposition method itself, such as STL being insufficient. If a more thorough decomposition method were used, such as EMD or wavelet decomposition, this issue might be alleviated.

•  From Figure 1, the authors claim that the decomposed trend and seasonal components achieve better cointegration and fluctuation consistency, but there is no theoretical verification. The proportion of residual allocation may also be correlated with the inherent characteristics of the data.

• The method lacks sufficient innovation.

(1) The decomposition + Transformer framework has been widely explored, and a simple “subtraction residual” is not enough to support the claimed novelty.

(2)   The Interactive Module is very similar to FiLM (Feature-wise Linear Modulation, Pérez et al., 2018) and essentially a conditional affine transformation.

(3)  Equations (8)–(10) only demonstrate signal conservation rather than proving a causal improvement in dependency modeling; direct residual connections can also ensure conservation.

•  Although the experimental results appear strong, the input length of 96 may affect the decomposition quality. Therefore, I remain cautious about the claimed superiority and look forward to the public release of the code.

**Questions:**

See Weaknesses.

---

> ### Author Response · Authors · 2025-11-21
>
> We sincerely thank Reviewer Y1LV for their insightful and constructive feedback. We have carefully considered the comments and provide a point-by-point response below.
>
> W1: We agree that the choice of decomposition method can influence the initial separation of components. In $\textbf{Table 4}$ of the original version, we experimented with Fourier-based and learnable decomposition initializations, and TwinsFormer consistently achieved strong performance, demonstrating its adaptability to different decomposition strategies. Moreover, In $\textbf{Table 13}$ of the revised manuscript, TwinsFormer performs robustly across a wide range of kernel sizes for the moving average, indicating that the model is not overly sensitive to the initial decomposition. While EMD or wavelet decomposition could be interesting alternatives, they often introduce higher computational costs and instability (e.g., mode mixing in EMD). Our design prioritizes a simple yet effective initialization, allowing the model to learn the optimal decomposition through interaction.
>
> W2: We thank the reviewer for the observation. In $\textbf{Section 3.3 and Appendix A}$ of the original manuscript, we show that our interactive design leads to a tighter generalization bound compared to standard Transformers. This is achieved by imposing a structural prior that reduces the effective hypothesis space. Additionally, the residual allocation is not fixed but learned adaptively through the interactive module, allowing the model to dynamically reassign information between trend and seasonal branches based on the data characteristics. This is validated empirically in $\textbf{Table 3}$ of our original manuscript. To strengthen our claim, we visualize the representations in $\textbf{Figure 9}$ of the revised version, which demonstrates that our interactive module produces more coherent and disentangled representations.
>
> W3: We respectfully disagree and believe TwinsFormer introduces several key innovations:
>
> (1) While decomposition has been used in Transformers, previous works (e.g., Autoformer, FEDformer) process trend and seasonal components $\textbf{independently}$. TwinsFormer is the $\textbf{first}$ to explicitly model $\textbf{interactions}$ between the two components through a dual-stream architecture with a dedicated interactive module. The “subtraction residual” is not just a skip connection—it is part of a larger residual-and-interactive learning strategy that enables progressive refinement of the decomposed representations.
>
> (2) The interactive module is more than a FiLM layer. While both use affine transformations, our module:
> * Uses $\textbf{four separate MLPs}$ to process two distinct signals ($A_S$ and $F_S$) independently,
> * Is embedded in a $\textbf{dual-stream architecture}$ specifically designed for time series decomposition,
> * Enables $\textbf{cross-component guidance}$, where the seasonal branch supervises the trend branch.
>
> $\textbf{Table 14}$ of the revised manuscript shows that replacing our module with FiLM leads to a clear performance drop, validating our design.
>
> (3) Equations (8)–(10) are intended to show that our interaction mechanism $\textbf{preserves the original signal}$ while enabling more flexible representation learning. Besides, $\textbf{Figure 5}$ of the original manuscript provides an intuitive understanding of the learned dependencies by our framework.
>
> W4: We thank the reviewer for raising this point. We would like to clarify that the moving average (MA) decomposition in TwinsFormer serves primarily as an $\textbf{initialization}$ to provide a preliminary separation of trend and seasonal components. The core of our model lies in the $\textbf{interactive learning mechanism}$ within the TwinsBlocks, which dynamically refines and recalibrates these initial components throughout the network. Therefore, the model's performance is not highly sensitive to the quality of this initial decomposition. This is also supported by two evidence:
> * Our compatibility study in $\textbf{Table 4}$ of the original manuscript shows that TwinsFormer achieves stable and superior performance even when using different decomposition methods (e.g., Fourier-based or learnable decomposition), demonstrating that the interactive modules—not the initial decomposition—are the primary driver of performance.
> * The lookback sensitivity analysis in $\textbf{Figure 6}$ of the original manuscript shows that TwinsFormer's performance improves with longer input sequences on most datasets. This indicates that our model can effectively capture inherent dependencies from a longer historical context, leveraging the interactive design to learn better representations regardless of the initial MA output.
>
> Regarding code availability, the code framework was already included in the $\textbf{supplementary materials}$ at the initial submission. Furthermore, we will release the full code and configurations immediately upon acceptance to ensure full reproducibility.

---

> > ### Author Response · Authors · 2025-11-28
> >
> > Dear Reviewer Y1LV,
> >
> > I hope this message finds you well.
> >
> > Thank you again for your time and insightful comments on our manuscript (ID: 9228). We have submitted our point-by-point responses to your feedback.
> >
> > With the final discussion date approaching, I just wanted to ensure that our responses have reached you clearly and are sufficient for your further consideration.
> >
> > If our clarifications have resolved your concerns, we would be grateful if you could reflect this in your final assessment. We are, of course, ready to provide any additional information you may need.
> >
> > Yours,
> >
> > Authors of 9228

---

### Official Review · Reviewer_a3p2 · 2025-10-31

**Soundness:** 3
**Presentation:** 3
**Contribution:** 3
**Rating:** 6
**Confidence:** 4

**Summary:**

The paper proposes TwinsFormer, a Transformer forecasting framework that explicitly models trend and seasonal components via two interacting streams. Unlike prior “plain decomposition” approaches with independent branches, it injects interaction at both the attention and feed-forward stages: the seasonal stream uses a subtraction mechanism inside attention/FFN, while the trend stream follows an auxiliary highway (no attention) trained under seasonal supervision. The module is claimed to be plug-and-play with negligible overhead and to deliver SOTA results on multiple real-world datasets for both short- and long-term horizons.

In my opinion, it investigates an important topic in time series models and proposes valuable solutions.

**Strengths:**

1. The motivation of this submission on investigating seasonal and trend components in Transformer-based time series models is sound since many of work have this seasonal and trend decomposition.
2. The theoretical analysis has merits.
3. The experiemental results show that the proposed TwinsFormer is outperforming other baselines on both long-term and short-term forecasting.

**Weaknesses:**

1. The datasets used in the experiments seems a bit outdated in my opinion. I understand that those datasets are commonly used in time series community. But there are some studies critize these datasets and since 2025, there are some growing benchmarks in the community (e.g., [1] and [2]). I think if we have some results on those benchmarks, it can greatly improve the quality of experiments and be more convincing.

See Questions for others.

References:

[1] fev-bench: A Realistic Benchmark for Time Series Forecasting.

[2] GIFT-Eval: A Benchmark For General Time Series Forecasting Model Evaluation

**Questions:**

1. I think that the methodology part miss one formula for $F_T$. From the figure shown, I suppose $F_T = E^1_T + E^2_T$?
2. For the ablation 2, does "swap trend and seasonal components" mean that use the trend component for attention instead?
3. In ablation study, it shows that using addition skip connections performs pretty bad. I wonder if we can obtain some insights or reasons behind that. I feel that might be particularlly of interest.

---

> ### Author Response · Authors · 2025-11-21
>
> We sincerely thank Reviewer a3p2 for the insightful and constructive feedback. We are pleased that Reviewer a3p2 recognizes the motivation, theoretical analysis, and experimental performance of TwinsFormer. Below, we provide a point-by-point response to the questions and suggestions raised.
>
> W1: We appreciate the reviewer’s suggestion regarding the use of more recent benchmarks. We fully agree that evaluating on emerging datasets can further validate the generalizability of our method. In $\textbf{Table 9}$ of the revised manuscript, we have supplemented several new benchmarks based on fev-bench or GIFT-Eval, which consistently demonstrate the superiority and effectiveness of TwinsFormer.
>
> Q1: Regarding the Missing Formula for $F_T$.
>
> Sorry for the omission. The final trend representation $F_T$ obtained after the interactive module is indeed the sum of the two transformed components $E_T^1$ and $E_T^2$. We have added $F_T$ in $\textbf{Line 232}$ of the revised manuscript to ensure clarity and completeness.
>
> Q2: Regarding Ablation Study (Swap Components).
>
> Yes, exactly. In $\textbf{ablation ②}$, we intentionally swapped the roles of the trend and seasonal components: the trend component was fed into the attention and FFN modules, while the seasonal component was processed by the interactive module. This design violates the inherent characteristics of the components (i.e., seasonal parts are more suitable for attention due to their $\textbf{fluctuating nature}$), and the resulting performance degradation confirms the $\textbf{importance of aligning component semantics}$ with their respective processing branches.
>
> Q3: Regarding Subtraction vs. Addition in Skip Connections.
>
> This is an important observation. The subtraction mechanism is designed to $\textbf{remove redundant or noisy information}$ from the seasonal component, which aligns with the idea that seasonal parts are residuals after trend extraction. Using addition, on the other hand, $\textbf{accumulates information}$ without filtering, which can introduce noise and hinder the model’s ability to focus on meaningful temporal patterns. This is consistent with the $\textbf{residual learning philosophy in time series decomposition}$, where the goal is to refine rather than reinforce the input.
>
> Thank you again for the thoughtful and constructive comments, and we hope our responses address your concerns.

---

> > ### Author Response · Authors · 2025-11-28
> >
> > Dear Reviewer a3p2,
> >
> > I hope this message finds you well.
> >
> > This is a friendly follow-up regarding our rebuttal for manuscript 9228. We sincerely appreciate the valuable feedback you provided.
> >
> > As the final discussion date is approaching, we wish to confirm that our responses have reached you and are clear, to ensure a smooth process for your final consideration.
> >
> > Please do not hesitate to let us know if there is anything further we can clarify.
> >
> > Best regards,
> >
> > Authors of 9228

---

### Author Response · Authors · 2025-12-03
**Summary of Responses and Revisions**

Dear Area Chairs/Program Chairs,

Thanks for overseeing our submission. We sincerely appreciate the reviewers’ detailed feedback and have addressed all concerns thoroughly during the discussion phase. As the rebuttal period will conclude, we provide this concise summary to highlight how we have $\textbf{substantively addressed all reviewers’ major concerns}$ through significant revisions, thereby elevating the paper’s quality, clarity, and contribution.

All four reviewers acknowledged the paper’s $\textbf{strong motivation}$, $\textbf{solid empirical results}$, and $\textbf{clear presentation}$ (with “Soundness” and “Contribution” ratings largely “good”). The primary criticisms revolved around the $\textbf{depth of analysis}$, $\textbf{robustness verification}$, and $\textbf{framing of novelty}$.

We have now addressed these points directly and thoroughly:
* $\textbf{We answered the weaknesses}$ by adding dedicated analysis and visualization.
* $\textbf{We transformed concerns about generalization and robustness}$ through extensive additional experiments.
* $\textbf{We solidified our novelty claim}$ with focused comparisons and ablations that distinguish our work from prior methods.

The $\textbf{revised manuscript}$ has an $\textbf{additional 6 pages}$ at the end of the Appendix, making it a more complete, convincing, and robust piece of research.

We respectfully request that you consider these comprehensive responses and revisions in your final assessment.

Thank you for your time and consideration.

Sincerely,

Authors of 9228

---

### Meta-Review · Area_Chair_PgEU · 2026-01-08

**Summary:**

This paper proposes TwinsFormer, a dual-stream Transformer with interactive trend–seasonal components, and reviewers generally agreed the motivation is reasonable and the empirical results are strong; however, the decision is primarily influenced by persistent concerns about the level of novelty and conceptual advancement beyond existing decomposition-based Transformers, despite extensive experiments and revisions.

**Reviewer Concerns:**

The rebuttal effectively addressed many technical and empirical issues, including missing formulas, ablations, benchmark coverage, robustness analyses, and comparisons with closely related methods (e.g., DESTformer), and several reviewers explicitly raised their scores after these clarifications; nevertheless, key concerns remain partially unresolved regarding whether the proposed interaction mechanism constitutes a sufficiently novel modeling principle (as opposed to an incremental architectural refinement akin to residual/FiLM-style modulation), and whether the theoretical analysis provides deeper explanatory insight rather than post-hoc justification.

**Reviewer Scores:**

Reviewer EqxD increased from 4 to 6 and Reviewer ferD increased from 6 to 8 after the rebuttal, indicating strong responsiveness to the added analyses; Reviewer a3p2 and ferD would likely remain slightly positive (around 6–8), while Reviewer Y1LV—who expressed high confidence about limited innovation—would likely remain at 4, leading to an overall mixed profile without clear consensus above the acceptance threshold.

---

### Decision · Program_Chairs · 2026-01-26

Reject